# CPPO: Continual Learning for Reinforcement Learning with Human Feedback

**Han Zhang**[1,2]**, Yu Lei**[2*]**, Lin Gui**[3]**, Min Yang**[4]**, Yulan He**[4]**, Hui Wang**[2]**, Ruifeng Xu**[1,2,5*]

[1] Harbin Institute of Technology (Shenzhen)
[2] Peng Cheng Laboratory
[3] King's College London
[4] Shenzhen Institutes of Advanced Technology
[5] Guangdong Provincial Key Laboratory of Novel Security Intelligence Technologies

## Abstract

The approach of Reinforcement Learning from Human Feedback (RLHF) is widely used for enhancing pre-trained Language Models (LM), enabling them to better align with human preferences. Existing RLHF-based LMs however require complete retraining whenever new queries or feedback are introduced, as human preferences may differ across different domains or topics. LM retraining is often impracticable in most real-world scenarios, due to the substantial time and computational costs involved, as well as data privacy concerns. To address this limitation, we propose **C**ontinual **P**roximal **P**olicy **O**ptimization (CPPO), a novel method that is able to continually align LM with dynamic human preferences. Specifically, CPPO adopts a weighting strategy to decide which samples should be utilized for enhancing policy learning and which should be used for solidifying past experiences. This seeks a good trade-off between policy learning and knowledge retention. Our experimental results show that CPPO outperforms strong Continuous learning (CL) baselines when it comes to consistently aligning with human preferences. Furthermore, compared to PPO, CPPO offers more efficient and stable learning in non-continual scenarios.

## 1 Introduction

Recent studies (Stiennon et al., 2020; Bai et al., 2022a; Ouyang et al., 2022) have shown that Reinforcement Learning from Human Feedback (RLHF) can significantly enhance language models by aligning them with human intention. RLHF uses human preferences as a reward signal to fine-tune language models with the Proximal Policy Optimization (PPO) algorithm. The RLHF-based model can effectively generate answers preferred by humans for tasks that lack standardized solutions, such as summarization(Stiennon et al., 2020), translation(Kreutzer et al., 2018), and dialogue(Jaques et al., 2020), without over-optimizing metrics such as ROUGE(Lin, 2004) or BLEU(Papineni et al., 2002).

In real-world applications, learning continuously changing human preferences is more practical than learning invariable human preferences. For example, the progression from the onset of the COVID-19 virus in human society to widespread infections and finally to achieving herd immunity has seen corresponding changes in government policies and human perspective. An AI agent that keeps pace with the times should exhibit behavior that aligns with current government policies and human understanding preferences at different stages, rather than remaining static.

However, traditional alignment methods(Stiennon et al., 2020; Ouyang et al., 2022), lack flexibility for continual learning (CL) of human preferences. Recent approach (Bai et al., 2022a) tackles these problems by periodically retraining the Preference Model (PM) and policy based on both new and historical data, it might be inefficient and impractical due to the involved concerns of computational cost and data privacy.

In this paper, we propose a more practical approach by enhancing RLHF with continual learning (CL), aiming to optimize two conflicting objectives: preserving old knowledge and acquiring new

---

*   Corresponding authors: Yu Lei (leiy01@pcl.ac.cn) and Ruifeng Xu (xuruifeng@hit.edu.cn).

knowledge (Rolnick et al., 2019). This leads to a long-standing challenge known as the *stability-plasticity*[1] *dilemma* (Abraham & Robins, 2005). Moreover, due to the vast action space (vocabulary) of LMs, the RLHF algorithms (e.g., PPO) usually suffer from the issues of inefficiency and instability during training (Ramamurthy et al., 2022). To tackle these challenges, we attempt to seek a good tradeoff between policy learning and knowledge retention with stable learning by designing a sample-wise weighting strategy over the rollout[2] samples. Our weighting strategy is motivated by the fact that *a desired policy should always generate high-reward results with high probabilities*.

Specifically, we first categorize the rollout samples into five types according to their rewards and generation probabilities, as shown in Figure 1. We then assign each rollout sample with a policy learning weight $\alpha$ and a knowledge retention weight $\beta$, in the following way. 1) For a high-performance sample, we assign a high $\alpha$ and a high $\beta$, in order to consolidate the knowledge of this sample. 2) For a high-variance or overfitting sample, we assign a high $\alpha$ and a low $\beta$, so as to learn more knowledge of this sample and force the new policy to be different from the old one in generating such a sample. 3) For a noisy sample, we assign a low $\alpha$ and a low $\beta$ to decrease its impact on learning. 4) For a normal sample, we make no changes.

Based on the above weighting strategy, we develop a novel PPO-based method, named continual proximal policy optimization (CPPO). CPPO implements the weighting strategy in two different ways: heuristic and learnable, resulting in two different CPPO methods (see Section 3.2 for details). The heuristic approach sets the weight with linear gain or decay according to strategy. The learnable approach converts the strategy into several inequality constraints and learns the best weight by optimizing the Lagrange function.

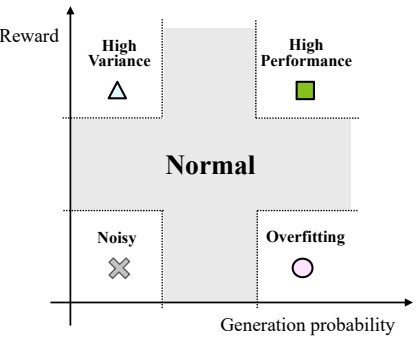

Experimental results on real-world summarization datasets demonstrate that our proposed CPPO methods significantly outperform the PPO re-training methods and the strong CL baselines, in both CL and non-CL settings (detailed in Appendix F). Furthermore, additional experiments in both settings verify the superior training stability of CPPO compared to the original PPO algorithm.

Figure 1: Five types of the rollout are utilized in our method.

## 2 PRELIMINARY AND TASK FORMULATION

PPO algorithm (Schulman et al., 2017) utilizes the clipped surrogate objective with a learned state-value function, and the entropy bonus (Mnih et al., 2016) is added to the original reward. The total objective is approximately maximized in each iteration step $i = 1, 2, ..., I$ (in the NLP scene, step-$i$ denotes the generation of the $i$-th token):

$$L_i^{CLIP+VF}(\theta) = \mathbb{E}_i[L_i^{CLIP}(\theta) - c \cdot L_i^{VF}(\theta)] \tag{1}$$

where $c$ is the coefficient, and $L_i^{VF}$ is a squared-error loss $(V_\theta(s_i) - V_i^{targ})^2$. The clipped policy learning objective is: $L_i^{CLIP}(\theta) = min(r_i(\theta) \cdot \mathbf{A}_i, clip(r_i(\theta), 1\pm\epsilon) \cdot \mathbf{A}_i)$, where $r_i(\theta) = \frac{\pi_\theta(a_i|s_i)}{\pi_{\theta_{old}}(a_i|s_i)}$ is the probability ratio, $\epsilon$ is the clip hyperparameter, $s_i$ is the $i$-th state, $\mathbf{A_i}$ is the truncated version of generalized advantage estimation.

**Task Formulation:** In this paper, we propose the task of continually learning human preferences under an offline continual learning setting(Biesialska et al., 2020). Formally. we consider a task sequence of $\mathbb{T} = \{\mathcal{T}_1, \mathcal{T}_2, ...\}$ to continually learn a policy model on the corresponding human

---

[1]In this context, stability refers to the retention of previously acquired knowledge, which is different from the training stability mentioned later. Plasticity, on the other hand, refers to the ability to adapt to new knowledge through policy learning.

[2]In the context of RLHF, a rollout, also known as a trajectory or episode, entails generating a response sequence, such as a summary, to a given conversation prompt, starting from a particular state (i.e. the initial prompt). The responses generated during the rollout are then used to update the policy network.

preference datasets $\mathbb{HF} = \{HF_1, HF_2, ...\}$ and prompt datasets $\mathbb{S} = \{S_1, S_2, ...\}$. For each task $\mathcal{T}_t(t = 1, 2, ...)$, the policy $\pi_t$ is initialized by $\pi_{t-1}$ and then is trained against the reward model $r_t$ , where the reward model $r_t$ is learned on $HF_t$. The initial policy $\pi_0$ is the SFT model, namely, $\pi_0 = \pi_{SFT}$. Let $x = (s, a)$ denote the prompt $s$ and answer $a$ pair. The final objective is to learn a policy model $\pi_\theta$ that maximizes the overall reward on all of the learned human preferences:

$$\max_\theta \Sigma_{t=1}^T \mathbb{E}_{s \sim S_t, a \sim \pi_\theta(\cdot|s)}[r_t(s, a)] \tag{2}$$

## 3  CONTINUAL PROXIMAL POLICY OPTIMIZATION

### 3.1  MOTIVATION AND THEORETICAL ANALYSIS

To optimize the objective 2 in the CL paradigm, the key is to balance the tradeoff between policy learning and knowledge retention, i.e., to learn a policy $\pi_t$ that not only fits current task $t$ but also retains the knowledge of previous tasks. This is typically accomplished by maximizing $\pi_t$'s average reward and meanwhile minimizing the difference between $\pi_t$ and $\pi_{t-1}$ by KL-based knowledge distillation (Kaplanis et al., 2019):

$$\max_\theta \mathbb{E}_{s \sim S_t, a \sim \pi_\theta(\cdot|s)}[r_t(s, a)] - \mathbb{E}_{s \in S_{t-1}} D_{\text{KL}}(P_{\pi_\theta}(a|s) \parallel P_{\pi_{t-1}}(a|s)) \tag{3}$$

where $P_{\pi_\theta}(a|s)$ denotes the probability that policy $\pi_\theta$ generates the answer $a$ to the prompt $s$. However, in the RLHF setting, we argue that a more effective way to achieve policy learning is to maximize the rewards of the results that $\pi_\theta$ has a high probability to generate. This is because LMs usually have a vast action space (vocabulary size) and adopt a sampling strategy such as beam search that favors high-probability generative results. For knowledge retention, on the other hand, it is more important to make $\pi_\theta$ retain $\pi_{t-1}$'s certain knowledge that generates high-reward outputs rather than all.

To accomplish the above ideas, we propose a theoretically desirable objective for continual RLHF at task $\mathcal{T}_t$:

$$\max_\theta \mathbb{E}_{(s,a) \in D_1} r_t(s, a) - \mathbb{E}_{(s,a) \in D_2} D_{\text{KL}}(P_{\pi_\theta}(a|s) \parallel P_{\pi_{t-1}}(a|s)) \tag{4}$$

where, $D_1 = \{(s, a)|s \sim S_t, a \sim \pi_\theta(\cdot|s), P_{\pi_\theta}(a|s) > \mu_a[P_{\pi_\theta}(a|s)] + k\sigma_a[P_{\pi_\theta}(a|s)]\}$ and $D_2 = \{(s, a)|s \sim S_{t-1}, a \sim \pi_{t-1}(\cdot|s), r_t(s, a) > \mu_a[r_t(s, a)] + k\sigma_a[r_t(s, a)]\}$ denote the sets of samples with high generation probability and high rewards, respectively. $\mu$ and $\sigma$ denote the mean and standard deviation respectively, and $k$ is a hyperparameter. It is important to note that here we use $r_t(s, a)$ instead of $r_{t-1}(s, a)$. Since the reward model is continually learneded, we assume $r_{t-1}(s, a) \approx r_t(s, a)$ when $s \in S_{t-1}$ and $a \sim \pi_\theta(\cdot|s)$. To simplify notation, the subsequent sections of the paper use $x$ instead of $(s, a)$.

The KL divergence term requires a significant amount of memory to store the probability distribution of each token across the vast vocabulary. To tackle this problem, we incorporate a low computational knowledge retention penalty term $L_i^{KR}(\theta) = (\log P_{\pi_\theta}(x_i) - \log P_{\pi_{t-1}}(x_i))^2$. We compute the L2 distance of the $\log$ generation probability of true tokens instead of the KL divergence of the entire vocabulary's probability distribution. We find the former is effective for knowledge retention and needs NOT to save the vocabulary's probability distribution in the memory[3].

We introduce $I_{D_1}(x)$ and $I_{D_2}(x)$ to denote the indicator functions of the sets of $D_1$ and $D_2$, respectively. By introducing the actor-critic version, the clipped ratio, and the entropy bonus, we claim that Eq.(4) can be written to (the derivation is detailed in Appendix Section B):

$$\begin{aligned} \mathbf{J}'(\theta) &= L_i^{I_{D_1} \cdot CLIP + I_{D_2} \cdot KR + VF}(\theta) \\ &= \mathbb{E}_i[I_{D_1}(x) \cdot L_i^{CLIP}(\theta) - I_{D_2}(x) \cdot L_i^{KR}(\theta) - c \cdot L_i^{VF}(\theta)] \end{aligned} \tag{5}$$

Compared with objective Eq. (1), Eq.(5) introduces the learning weights $I_{D_1}(x)$, $I_{D_2}(x)$, and the $L_i^{KR}$ loss. Unfortunately, it is still impractical to directly optimize the objective, since the training

---

[3]In our task, the reference model generates 512 summaries (max 50 tokens) in one rollout. The vocabulary size is nearly 5e+4. If we use FP16 to save the logits or proability tensor, it takes about 512*50*5e4*2 Bit/1e9 = 1.28GB of memory. However, computing $L^{KR}$ only needs to save the probability of true tokens, which takes only 512*50*2 Bit/1e9 = 2.56E-05GB of memory.

samples in $D_1$ and $D_2$ are seldom as indicated by the *Cantelli Inequation*[4] $P(\mathbf{X} > \mu[\mathbf{X}] + k\sigma[\mathbf{X}]) < 1/(1 + k^2)$. To make Eq.(5) easy to optimize, we generalize the indicator functions $I_{D_1}(x)$ and $I_{D_2}(x)$ to positive real-valued functions $\alpha(x)$ and $\beta(x)$, *which gives each sample a non-zero learning weight*.

## 3.2 WEIGHTING STRATEGY

Our method utilizes sample-wise balance weights $\alpha(x)$ and $\beta(x)$ to regulate the policy learning and knowledge retention processes, aiming to find a balance between knowledge retention and policy learning. The final objective is:

$$
\begin{aligned}
\mathbf{J}(\theta) &= L_i^{\alpha \cdot CLIP + \beta \cdot KR + VF}(\theta) \\
&= \mathbb{E}_i[\alpha(x)L_i^{CLIP}(\theta) - \beta(x)L_i^{KR}(\theta) - c \cdot L_i^{VF}(\theta)]
\end{aligned}
\tag{6}
$$

for task $t = 1, 2, ..., T$. Next, we propose a weighting strategy for balancing policy learning and knowledge retention.

### 3.2.1 BALANCING POLICY LEARNING AND KNOWLEDGE RETENTION

To simplify the expression, we define the operator $F[\cdot] = \mu[\cdot] - k\sigma[\cdot]$ and operator $G[\cdot] = \mu[\cdot] + k\sigma[\cdot]$. As shown in Figure 1 and Table 1, we classify the rollout samples into 5 rollout types based on the joint distribution of $(\mathbf{P}_{\pi_\theta}(x), \mathbf{R}(x))$. If $\mathbf{P}_{\pi_\theta}(x)$ or $\mathbf{R}(x)$ is outside the discriminant interval $(F[\cdot], G[\cdot])$, it is considered as high or low. Now, we detail each rollout type and corresponding weight strategy.

**High-performance sample:** If both $\mathbf{P}_{\pi_\theta}(x)$ and $\mathbf{R}(x)$ are high, it indicates that the old policy has high confidence to generate $x$ which gets a high reward, implying that it is already performing well. In this case, we ask the new policy to enhance both policy learning and knowledge retention.

**Overfitting sample:** A high $\mathbf{P}_{\pi_\theta}(x)$ with a low $\mathbf{R}(x)$ indicates that the old policy is likely overfitting (due to high probability) to the biased sample (due to low reward score). We aim to reduce the generation probability of the biased sample $x$, which can be achieved through policy learning. However, knowledge retention will maintain the high probability of the biased sample $x$. Therefore, we enhance policy learning and slow down knowledge retention.

Table 1: The determining condition of rollout type and corresponding weight strategy to balance policy learning and knowledge retention. We monitor the generating probability $\mathbf{P}_{\pi_\theta}(x)$ and the corresponding reward score $\mathbf{R}(x)$. The rollout type of sample $x$ depends on whether the $\mathbf{P}_{\pi_\theta}(x)$ and $\mathbf{R}(x)$ fall in or outside the discriminant interval $(F[\cdot], G[\cdot])$.

| ID | Rollout Type | Determining Condition | | Weight Strategy | |
|---|---|---|---|---|---|
| $r_1$ | High-performance | $\mathbf{P}_{\pi_\theta}(x) \geq G[\mathbf{P}_{\pi_\theta}]$ | $\mathbf{R}(x) \geq G[\mathbf{R}]$ | $\alpha(x) \uparrow$ | $\beta(x) \uparrow$ |
| $r_2$ | Overfitting | $\mathbf{P}_{\pi_\theta}(x) \geq G[\mathbf{P}_{\pi_\theta}]$ | $\mathbf{R}(x) \leq F[\mathbf{R}]$ | $\alpha(x) \uparrow$ | $\beta(x) \downarrow$ |
| $r_3$ | High-variance | $\mathbf{P}_{\pi_\theta}(x) \leq F[\mathbf{P}_{\pi_\theta}]$ | $\mathbf{R}(x) \geq G[\mathbf{R}]$ | $\alpha(x) \uparrow$ | $\beta(x) \downarrow$ |
| $r_4$ | Noisy | $\mathbf{P}_{\pi_\theta}(x) \leq F[\mathbf{P}_{\pi_\theta}]$ | $\mathbf{R}(x) \leq F[\mathbf{R}]$ | $\alpha(x) \downarrow$ | $\beta(x) \downarrow$ |
| $r_5$ | Normal | $\mathbf{P}_{\pi_\theta}(x)$ or $\mathbf{R}(x) \in (F, G)$ | | $-$ | $-$ |

**High-variance sample:** If $\mathbf{P}_{\pi_\theta}(x)$ is low while $\mathbf{R}(x)$ is high, it suggests that sample $x$ has high variance. Due to the low $\mathbf{P}_{\pi_\theta}(x)$, the likelihood of generating $x$ next time is low. To achieve stable (low variance) performance, we aim to increase the generation probability of sample $x$, which can be accomplished through policy learning. However, knowledge retention will maintain a low generation probability. Therefore, we enhance policy learning and slow down knowledge retention.

**Noisy sample:** If both $\mathbf{P}_{\pi_\theta}(x)$ and $\mathbf{R}(x)$ are low, sample $x$ is considered noisy data which may lead to overoptimization against the PM (Gao et al., 2022). Therefore, we slow down both knowledge retention and policy learning.

**Normal sample:** If at least one of $\mathbf{P}_{\pi_\theta}(x)$ and $\mathbf{R}(x)$ falls within the discriminant interval, we consider it a normal condition and do not alter the learning process.

---

[4]Cantelli's inequality (also called the Chebyshev-Cantelli inequality and the one-sided Chebyshev inequality) is a version of Chebyshev's inequality for one-sided tail bounds.

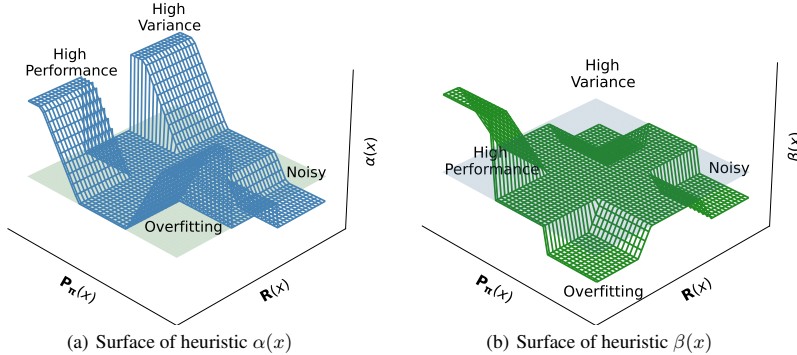

(a) Surface of heuristic $\alpha(x)$        (b) Surface of heuristic $\beta(x)$

Figure 2: The surfaces of heuristic weights. The weights are equal to 1 when rollout samples fall in the normal zone.

### 3.2.2 How to determine balance weights?

The above weight strategies constitute several inequality constraints of $\alpha(x)$ and $\beta(x)$, shown in Table 2. Determining balance weights requires finding a feasible solution that satisfies those constraints. We provide two methods to determine balance weights including the heuristic weight method and the learnable weight method.

Table 2: The constraint of weights and heuristic weights.

| ID | Constraint of $\alpha(x)$ | Constraint of $\beta(x)$ | Heuristic $\alpha(x)$ | Heuristic $\beta(x)$ |
|---|---|---|---|---|
| $r_1$ | $\alpha(x_{r_5}) - \alpha(x_{r_1}) < 0$ | $\beta(x_{r_5}) - \beta(x_{r_1}) < 0$ | $\min(ub, \frac{P_{\pi_\theta}(x) - \mu[P_{\pi_\theta}]}{k\sigma[\pi_\theta]})$ | $\min(ub, \frac{\mathbf{R}(x) - \mu[\mathbf{R}]}{k\sigma[\mathbf{R}]})$ |
| $r_2$ | $\alpha(x_{r_5}) - \alpha(x_{r_2}) < 0$ | $\beta(x_{r_2}) - \beta(x_{r_5}) < 0$ | $\min(ub, \frac{P_{\pi_\theta}(x) - \mu[P_{\pi_\theta}]}{k\sigma[\pi_\theta]})$ | $\max(lb, 2 + \frac{\mathbf{R}(x) - \mu[\mathbf{R}]}{k\sigma[\mathbf{R}]})$ |
| $r_3$ | $\alpha(x_{r_5}) - \alpha(x_{r_3}) < 0$ | $\beta(x_{r_3}) - \beta(x_{r_5}) < 0$ | $\min(ub, \frac{P_{\pi_\theta}(x) - \mu[P_{\pi_\theta}]}{k\sigma[\pi_\theta]})$ | $\max(lb, 2 + \frac{\mathbf{R}(x) - \mu[\mathbf{R}]}{k\sigma[\mathbf{R}]})$ |
| $r_4$ | $\alpha(x_{r_4}) - \alpha(x_{r_5}) < 0$ | $\beta(x_{r_4}) - \beta(x_{r_5}) < 0$ | $\max(lb, 2 + \frac{P_{\pi_\theta}(x) - \mu[P_{\pi_\theta}]}{k\sigma[\pi_\theta]})$ | $\max(lb, 2 + \frac{\mathbf{R}(x) - \mu[\mathbf{R}]}{k\sigma[\mathbf{R}]})$ |
| $r_5$ | — | — | 1 | 1 |
| All | $\mathbb{E}_{x \sim \pi_{t-1}}[\alpha(x)] = 1$ | $\mathbb{E}_{x \sim \pi_{t-1}}[\beta(x)] = 1$ | — | — |

**Heuristic $\alpha(x)$ and $\beta(x)$:** If $\mathbf{P}_{\pi_\theta}(x)$ or $\mathbf{R}(x)$ fall within the discriminant interval, the balance weights are set to 1. If they are further away from the discriminant interval, the weights will linearly increase or decrease (depending on the rollout type). We can plot the surfaces of $\alpha(x)$ and $\beta(x)$ in 3D coordinate systems, as shown in Figure 2. The heuristic weights $\alpha(x)$ and $\beta(x)$ for a given sample $x$ can be calculated by the formula presented in Table 2.

**Learnable $\alpha(x)$ and $\beta(x)$:** Heuristic $\alpha(x)$ and $\beta(x)$ lack enough adaptation ability to the dynamic learning process. Hence, we propose the learnable balance weights to automatically balance policy learning and knowledge retention. We learn 2N parameters for each rollout batch in which the LM generates N responses, the 2N parameters can be discarded before the next rollout batch.

Our goal is to find a set of weights that satisfy the constraints in Table 2. Unlike the typical optimization problem solved by the Lagrange Multiplier method, we do not need to minimize an additional objective function. It should be noted that the optimization objective of CPPO in Eq.6 is not directly optimized using the Lagrange Multiplier method.

We employ a more straightforward strategy. We construct an unconstrained optimization objective by adding all the terms on the left side of the inequalities (in Table 2) together:

$$\begin{aligned}
\mathbf{L}_{coef}(\phi) = \mathbb{E}_{x \sim \pi_{t-1}}[&(\alpha_\phi(x) - 1)^2 + (\beta_\phi(x) - 1)^2] + \tau(\alpha(x_{r_5}) - \alpha(x_{r_1}) + \beta(x_{r_5}) - \beta(x_{r_1}) \\
&+ \alpha(x_{r_5}) - \alpha(x_{r_2}) + \beta(x_{r_2}) - \beta(x_{r_5}) + \alpha(x_{r_5}) - \alpha(x_{r_3}) + \beta(x_{r_3}) - \beta(x_{r_5}) \\
&+ \alpha(x_{r_4}) - \alpha(x_{r_5}) + \beta(x_{r_4}) - \beta(x_{r_5}))
\end{aligned} \quad (7)$$

where, $\alpha(x) = (ub - lb) \cdot sig(\phi_x^1) + lb$, $\beta(x) = (ub - lb) \cdot sig(\phi_x^2) + lb$, and $sig$ is sigmoid function, $lb$ and $ub$ are lower and upper bound of $\alpha(x)$ and $\beta(x)$. We directly optimize Eq. 7 using SGD to find

a set of weights that satisfy the constraints. We set multiplier $\tau$ as a hyperparameter, and $\tau = 0.1$ is selected from {0.01, 0.1, 0.5, 1.0}. For more hyperparameter sensitivity analysis experiments, please refer to Appendix E.1. We found this simple idea is highly effective in our scenario. In Appendix E.2, we analyze the time and memory required for SGD to find feasible solutions and found that it does NOT significantly increase the overall training time and memory.

# 4 EXPERIMENTS

We assess the performance of CPPO and baseline methods in the domain incremental learning (DIL) summary task. We also evaluate CPPO on non-continual learning tasks (Appendix Section F).

## 4.1 THE EXPERIMENTAL CONFIGURATION FOR CONTINUAL LEARNING FROM HUMAN PREFERENCES

**Dataset and split**: In accordance with previous research (Stiennon et al., 2020), we evaluate our method using the Reddit TL;DR (Völske et al., 2017) dataset for summarization. We use the human preference data provided by CarperAI[5]. To the best of our knowledge, there are limited benchmark datasets proposed for evaluating continual RLHF methods. Consequently, we divide the Reddit TL;DR dataset based on domains into two parts, which are outlined in Table 3. Each part corresponds to a distinct alignment task.

**Experiment settings**:
We evaluate CPPO under the DIL setting with two tasks, and the historical data is assumed inaccessible. This scenario is typical in real-world applications, such as developers continually learning an open-Source

Table 3: The dataset is utilized for continual learning. The human feedback data is used for training the reward model. The post (prompt) and summary (label) of Reddit TL;DR are used for SFT. The domain of "r / others" includes 28 categories, such as books, travel, and cooking. It's worth noting that the summary (label) data is not used in the reinforcement learning (RL) process.

| Task ID | Data | Data split | Train | Valid | Test | Domain |
|---------|------|-----------|-------|-------|------|--------|
| **task-1** | **Human Feedback** | part-1 | 52243 | - | 45148 | r / relationships |
| | **Reddit TL;DR** | part-1 | 63324 | 3462 | 3539 | r / relationships |
| **task-2** | **Human Feedback** | part-2 | 40291 | - | 38481 | r / others |
| | **Reddit TL;DR** | part-2 | 53398 | 2985 | 3014 | r / others |

RLHF model like vicuna(Chiang et al., 2023) in a special domain (e.g., game) without permission to access the pre-training corpus. For each task, we employ a 1.3B gpt2-xl (Radford et al., 2019) model with a value head as the reward model (RM). The RM is continually trained for 5 epochs on each task using the MAS(Aljundi et al., 2018) method. Since the policy is prone to over-optimize against the PM (Gao et al., 2022), we train a 6.7B gptj (Wang & Komatsuzaki, 2021) model as the reference PM (rPM) to measure the performance of alignment. The rPM is trained on entire human preferences data. We conduct experiments to evaluate the RM trained with and without MAS through accuracy and forgetting ratio (Chaudhry et al., 2018) (FR) of accuracy. The evaluation results of RM and rPM are shown in Table 4. The accuracy is computed by counting the percentage of the reward scores of human-preferred responses that are higher than the reward scores of human-NOT-preferred responses(Yuan et al., 2023). We initialize the SFT model from gpt2-s and train it on the Reddit TL;DR part-1 for 5 epochs. However, we do not perform the SFT process in task-2 as we observe no significant effects on performance.

**Metrics**: We use the forgetting radio (Chaudhry et al., 2018) of the ROUGE and reference PM score to measure the extent to which the old policy is forgotten. Notably, we consider the alignment tax (Ouyang et al., 2022) as part of forgetting since it arises when the SFT model

Table 4: The evaluation results of RMs and rPM.

| Reward Model | Acc($HF_1^{test}$) | Acc($HF_2^{test}$) | FR |
|--------------|------------------|------------------|-----|
| $RM_1$ | 0.7441 | - | - |
| $RM_2$ w MAS | 0.7203 | 0.7482 | 0.024 |
| $RM_2$ w/o MAS | 0.6971 | 0.7496 | 0.047 |
| rPM | 0.7624 | 0.7592 | - |

learns human preferences during the RL step. After learning all tasks, we evaluate the models on the entire test set using both reference PM score and ROUGE score. Table 5 presents the metrics used to

---

[5]URL: https://huggingface.co/datasets/CarperAI/openai_summarize_comparisons

evaluate each task, as well as the final evaluation metric. A well-performing model is expected to achieve high scores in both the reference PM and ROUGE metrics.

Table 5: Metrics for our tasks. $\mathbb{D}_i^{test}(i = 1, 2)$ denote the test data of Reddit TL;DR data part-i, and $rPM(M_i, \mathbb{D}_i^{test})(i = 1, 2)$ denote the reference PM score of model $M_i$ on dataset $\mathbb{D}_i^{test}$.

|  | Metric | Definition |
|---|---|---|
| Task-1 | reference PM Score on Task-1 (rPMS$_1$, ↑) | $rPM(M_1, \mathbb{D}_1^{test})$ |
| Task-1 | Alignment Tax (AT, ↓) | $Rouge(M_{SFT}, \mathbb{D}_1^{test}) - Rouge(M_1, \mathbb{D}_1^{test})$ |
| Task-2 | reference PM Score on Task-2 (rPMS$_2$, ↑) | $rPM(M_2, \mathbb{D}_2^{test})$ |
| Task-2 | Score Forgetting Ratio (SFR, ↓) | $rPM(M_1, \mathbb{D}_1^{test}) - rPM(M_2, \mathbb{D}_1^{test})$ |
| Final eval | reference PM Score on entire test data (rPMS, ↑) | $rPM(M_2, \mathbb{D}_1^{test} \cup \mathbb{D}_2^{test})$ |

## 4.2 RESULTS OF CONTINUAL LEARNING FROM HUMAN PREFERENCES

Table 6 shows the results of continual learning from human preferences on the summary task. We observe that CL methods, such as EWC (Kirkpatrick et al., 2017) regularization or policy consolidation (Kaplanis et al., 2019) can improve the training stability of the PPO method, thereby ensuring that the policy does not change too much with every policy gradient step. This leads to improved rPMS. Our method outperforms CL baselines by achieving the most significant enhancement in policy learning (rPMS) and possessing Backward Transfer (BWT) (Lopez-Paz & Ranzato, 2017) capability (negative SFR). This is because our learning strategy is sample-adaptive and balances policy learning and knowledge retention. Additionally, CPPO performs better than Iterated RLHF because PPO is not stable enough in the learning process. We observed that during PPO training, the KL divergence and value prediction errors tend to increase suddenly, as discussed in Section 4.4.

Table 6: The main results of continual alignment on TL; DR dataset. For PPO (In order)$^*$, we directly finetune the RM$_1$ on the novel data to obtain RM$_2$, without using MAS regularization; and we directly train the policy model $M_{\pi_1}$ against RM$_2$ to obtain $M_{\pi_2}$. For the Iterated RLHF†(PPO), we retrain the RM$_2$ and policy model $M_{\pi_2}$ on the combination of the Task-1 and Task-2 corpus. Methods in italics are trained against the continually learned (by MAS) reward models. Details of the implementation can be found in Appendix G.

| Method | Task-1 ($M_{\pi_1}$) | | | Task-2 ($M_{\pi_2}$) | | | Final eval ($M_{\pi_2}$) | |
|---|---|---|---|---|---|---|---|---|
| | rPMS$_1$ (↑) | rouge (↑) | AT (↓) | rPMS$_2$ (↑) | rouge (↑) | SFR (↓) | rPMS (↑) | rouge (↑) |
| **Human** | 2.958 | – | – | 2.805 | – | – | 2.903 | – |
| **ChatGPT** | 3.298 | 0.197 | – | 3.189 | 0.191 | – | 3.242 | 0.193 |
| **SFT (In order)** | 1.499 ±0.130 | **0.248** ±0.006 | – | 1.543 ±0.067 | **0.237** ±0.007 | – | 1.498 ±0.051 | **0.237** ±0.009 |
| **SFT (multi-tasks)** | 1.524 ±0.041 | 0.254 ±0.011 | – | 1.536 ±0.092 | 0.234 ±0.009 | – | 1.505 ±0.011 | 0.236 ±0.008 |
| **PPO (In order)**$^*$ | 2.629 ±0.183 | 0.196 ±0.050 | 0.052 ±0.044 | 2.546 ±0.201 | 0.151 ±0.022 | 0.144 ±0.024 | 2.502 ±0.242 | 0.186 ±0.016 |
| **Iterated RLHF†** | 2.629 ±0.183 | 0.196 ±0.050 | 0.052 ±0.044 | 2.732 ±0.163 | 0.211 ±0.011 | 0.061 ±0.018 | 2.666 ±0.124 | 0.200 ±0.010 |
| *PPO* | *2.629 ±0.183* | *0.196 ±0.050* | *0.052 ±0.044* | *2.687 ±0.126* | *0.184 ±0.017* | *0.080 ±0.017* | *2.612 ±0.191* | *0.188 ±0.013* |
| *PPO+OnlineL2 Reg* | *2.758 ±0.121* | *0.206 ±0.042* | *0.042 ±0.042* | *2.701 ±0.205* | *0.180 ±0.012* | *0.062 ±0.013* | *2.700 ±0.114* | *0.196 ±0.011* |
| *PPO+EWC (Kirkpatrick et al., 2017)* | *2.833 ±0.122* | *0.201 ±0.043* | *0.047 ±0.039* | *2.823 ±0.192* | *0.175 ±0.022* | *0.040 ±0.015* | *2.801 ±0.202* | *0.196 ±0.023* |
| *PPO+MAS (Aljundi et al., 2018)* | *2.712 ±0.132* | *0.211 ±0.051* | *0.034 ±0.037* | *2.726 ±0.189* | *0.157 ±0.021* | *0.039 ±0.020* | *2.714 ±0.167* | *0.179 ±0.011* |
| *PPO+LwF (Li & Hoiem, 2018)* | *2.822 ±0.126* | *0.197 ±0.051* | *0.048 ±0.050* | *2.832 ±0.179* | *0.169 ±0.036* | *0.030 ±0.019* | *2.824 ±0.192* | *0.179 ±0.019* |
| *PPO+TFCL (Aljundi et al., 2019)* | *2.867 ±0.109* | *0.202 ±0.039* | *0.043 ±0.046* | *2.864 ±0.169* | *0.169 ±0.020* | *0.054 ±0.022* | *2.842 ±0.211* | *0.178 ±0.014* |
| *PC (Kaplanis et al., 2019)* | *2.692 ±0.117* | *0.209 ±0.048* | *0.036 ±0.055* | *2.723 ±0.195* | *0.165 ±0.019* | *0.047 ±0.017* | *2.703 ±0.191* | *0.187 ±0.016* |
| *HN-PPO (Schöpf et al., 2022)* | *2.859 ±0.105* | *0.212 ±0.034* | *0.036 ±0.042* | *2.868 ±0.132* | *0.171 ±0.017* | *0.028 ±0.029* | *2.846 ±0.177* | *0.201 ±0.011* |
| *NLPO (Ramamurthy et al., 2022)* | *2.784 ±0.102* | *0.185 ±0.041* | *0.060 ±0.050* | *2.796 ±0.116* | *0.172 ±0.021* | *0.012 ±0.012* | *2.799 ±0.146* | *0.181 ±0.022* |
| *CPPO (Heuristic)* | *3.020 ±0.137* | *0.213 ±0.024* | *0.035 ±0.023* | *2.978 ±0.113* | *0.174 ±0.019* | ***-0.164 ±0.009*** | *3.099 ±0.153* | *0.179 ±0.016* |
| *CPPO (Learn)* | ***3.180 ±0.154*** | *0.220 ±0.040* | ***0.028 ±0.042*** | ***3.085 ±0.134*** | *0.164 ±0.024* | *-0.161 ±0.008* | ***3.207 ±0.113*** | *0.179 ±0.008* |

## 4.3 ABLATION STUDY

We conduct an ablation study on our proposed CPPO method. To analyze the effect of the balance weights, we conduct experiments by setting either $\alpha(x)$ or $\beta(x)$ to 1. To analyze the effect of the knowledge retention penalty, we set $\beta(x) \equiv 0$. The training curves of different weights are shown in Figure 3, and the evaluation results are presented in Table 7. We observe that the training process becomes unstable when setting $\beta(x)$ to 0. When setting $\alpha(x)$ to 1 reduces the rPMS, the noisy samples are learned together with normal samples without distinction, hence the reward increase is slower than CPPO. When setting $\beta(x)$ to 1 increases the SFR, the overfitting samples, high-variance samples, and noisy samples are consolidated the in the knowledge retention process, hence the final reward value is lower than CPPO. The above experiments indicate that the sample-wise balance weights are helpful for both policy learning and knowledge retention.

Table 7: Ablation study. PPO is a special case of CPPO ($^*\alpha \equiv 1, \beta \equiv 0$).

| Method | Task-1 | | | Task-2 | | |
|---|---|---|---|---|---|---|
| | rPMS$_1$ ($\uparrow$) | rouge ($\uparrow$) | AT ($\downarrow$) | rPMS$_2$ ($\uparrow$) | rouge ($\uparrow$) | SFR ($\downarrow$) |
| CPPO / **H**euristic | 3.020 ±0.137 | 0.213 ±0.024 | 0.035 ±0.023 | 2.978 ±0.113 | 0.174 ±0.019 | **-0.164** ±0.009 |
| CPPO / **L**earn | **3.180** ±0.154 | **0.220** ±0.040 | **0.028** ±0.042 | **3.085** ±0.134 | 0.164 ±0.024 | -0.161 ±0.008 |
| PPO / $\alpha \equiv 1, \beta \equiv 0$ | 2.629 ±0.183 | 0.196 ±0.050 | 0.052 ±0.044 | 2.687 ±0.126 | 0.184 ±0.017 | 0.080 ±0.017 |
| CPPO / $\alpha \equiv 1$ | 2.837 ±0.124 | 0.196 ±0.029 | 0.047 ±0.041 | 2.745 ±0.121 | 0.169 ±0.020 | -0.031 ±0.010 |
| CPPO / $\beta \equiv 1$ | 2.476 ±0.117 | 0.185 ±0.021 | 0.063 ±0.025 | 2.520 ±0.119 | **0.186** ±0.017 | 0.051 ±0.009 |
| CPPO / $\beta \equiv 0$ | 2.012 ±0.186 | 0.209 ±0.022 | 0.038 ±0.045 | 2.436 ±0.141 | 0.174 ±0.021 | 0.142 ±0.015 |

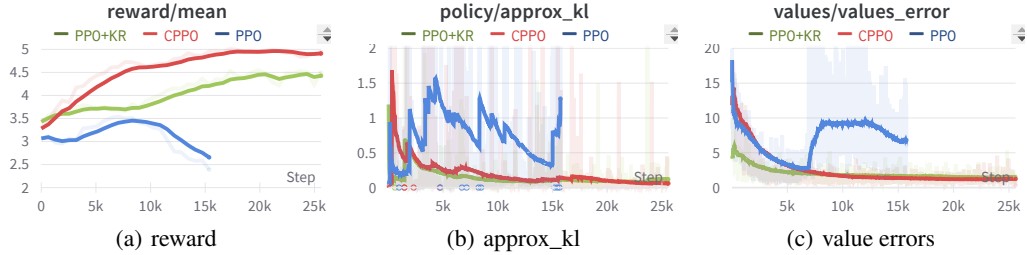

(a) reward      (b) approx_kl      (c) value errors

Figure 3: The curves of different weights in task-1. The knowledge retention weights penalty can improve the training stability of the PPO algorithm. However, setting $\beta(x) \equiv 1$ slows down the increase of the reward compared with CPPO. On the other hand, the policy learning weights $\alpha(x)$ can boost the increase of the reward compared with $\alpha(x) \equiv 1$.

## 4.4 STABILITY ANALYSIS

In this section, we analyze the stability of the CPPO, PPO, and PPO with the knowledge retention penalty. Previous work (Bai et al., 2022a) argues that small models are more prone to be unstable in PPO training. However, we find that CPPO can learn stably without the need for invalid-action masking(Ramamurthy et al., 2022), even with small models. As shown in Figure 4, the vanilla PPO performers unstably on the new data distribution. PPO with a knowledge retention penalty is more stable than PPO, but policy learning is slow. CPPO gets fast convergence on reward score and shows stable performance on the KL divergence and value prediction. This is because the sample-wise learning strategy of CPPO restricts the learning of noisy samples.

(a) reward      (b) approx_kl      (c) value errors

Figure 4: Training process of Task-2. The PPO algorithm is unstable at 7k steps and is unable to continuously increase the reward score.

## 4.5 HUMAN EVALUATION ON REDDIT TL;DR

We train two gpt2-xl models using CPPO and PPO, respectively, and compare their summaries with those generated by humans and ChatGPT using a Likert scale(Likert, 1932). The results are shown in Table 8. During the human evaluation, we observe that ChatGPT tends to generate longer summaries than humans and our models, but its performance remains stable across the test samples.

Although humans provide the best summaries, they still made mistakes, such as obfuscating important details. Our model achieves comparable performance with ChatGPT but still makes mistakes that the

small model often makes, such as repeating words and sentences. Due to the training inefficiency and instability, the performance of gpt2-xl trained by PPO is not satisfactory.

## 5 RELATED WORK

### 5.1 REINFORCEMENT LEARNING FROM HUMAN OR AI FEEDBACKS

Learning from human preferences has been studied in the game field (Bradley Knox & Stone, 2008) and has recently been introduced into the NLP domain, such as dialogue systems (Li et al., 2023; Zhao et al., 2023; 2024). Previous work (Stiennon et al., 2020) utilizes the PPO algorithm to fine-tune a language model (LM) for summarization and demonstrates that RLHF can improve the LM's generalization ability, which serves as the technology prototype for InstructGPT (Ouyang

Table 8: Human evaluation on 100 posts from the Reddit TL;DR.

| Method | Likert score | Improve | p-value |
|---------|--------------|---------|---------|
| PPO | $4.370_{\pm 1.180}$ | - | - |
| CPPO | $4.730_{\pm 1.231}$ | 8.23% | 0.037 |
| ChatGPT | $4.760_{\pm 1.011}$ | 8.92% | 0.013 |
| Human | $4.900_{\pm 1.034}$ | 12.13% | 0.001 |

et al., 2022) and ChatGPT. Learning LMs from feedback can be divided into two categories: human or AI feedback. Recent works such as HH-RLHF (Bai et al., 2022a) and InstructGPT (Ouyang et al., 2022) collect human preferences to train a reward model and learn a policy through it. ILF (Scheurer et al., 2023) proposes to learn from natural language feedback, which provides more information per human evaluation. Since human annotation can be expensive, learning from AI feedback (RLAIF) (Bai et al., 2022b; Perez et al., 2022; Ganguli et al., 2022) is proposed, but current methods are only effective for reducing harmless outputs, while helpful outputs still require human feedback.

### 5.2 CONTINUAL LEARNING

Within the realm of continual learning, several noteworthy methodologies emerge, encompassing the *regularization-based approach, replay-based techniques, optimization-based strategies, representation-based methodologies*, and *architecture-based innovations* (Wang et al., 2023).

The Regularization-Based Approach (Kirkpatrick et al., 2017; Aljundi et al., 2018; Chaudhry et al., 2018; Li & Hoiem, 2018; Castro et al., 2018) orchestrates the introduction of explicit regularization terms, thereby striving to strike a harmonious balance between the acquisition of new skills and the retention of past knowledge. *The Replay-Based Approach* aims to preserve and reuse past experiences to enhance model performance, which includes *experience replay*(Lin, 1992), *generative replay or pseudo-rehearsal* (Sun et al., 2020) and *feature replay*(Liu et al., 2020). *The Optimization-Based Approach* navigates the terrain of continual learning through explicit design and manipulation of optimization programs. This includes techniques such as *gradient projection*(Lopez-Paz & Ranzato, 2017), and *meta-learning*(Javed & White, 2019). *The Representation-Based Approach* leverages the strengths of self-supervised learning (SSL)(Gallardo et al., 2021) and large-scale pre-training(Mehta et al., 2022) to enhance the quality of representations at both the initialization and continual learning stages. *The Architecture-Based Approach* addresses inter-task interference by fashioning task-specific parameters. This approach can be dissected into three distinct paradigms: *parameter allocation*(Serra et al., 2018), *model decomposition*(Ebrahimi et al., 2020), and *modular networks*(Rusu et al., 2016).

## 6 CONCLUSION

In this work, we propose CPPO, which utilizes learning weights to balance policy learning and knowledge retention, with the aim of improving the PPO algorithm for continual learning from human preferences. CPPO is a task-agnostic and model-agnostic method that does not significantly increase the time and space complexity of PPO. We evaluate CPPO on both the DIL task and three non-continual tasks and show that it outperforms strong continual learning baselines when continually aligning with human preferences. Additionally, CPPO improves the learning efficiency and training stability of PPO. Our experiments demonstrate the potential of our approach for efficient and stable continual learning from human preferences, which can have applications in various domains and tasks.

## ACKNOWLEDGEMENTS

We thank the anonymous reviewers for their valuable suggestions to improve the quality of this work, and we express our sincere gratitude to Dr. Bin Liang for his invaluable guidance and constructive feedback throughout the preparation of this manuscript. This research was supported in part by the National Key Research and Development Program of China (2021ZD0112905), the Major Key Proiect of PCL (PCL2023A09-4), the National Natural Science Foundation of China (62176076), the Guangdong Provincial Key Laboratory of Novel Security Intelligence Technologies(2022B1212010005), Natural Science Foundation of Guangdong (2023A1515012922), Shenzhen Foundational Research Funding (JCYJ20220818102415032) and the UK Engineering and Physical Sciences Research Council (EP/X019063/1).

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

# A NOTATIONS

All of the notations used in this paper and their corresponding meanings are listed in Table 9.

Table 9: Notations used in this paper, *italic* font denotes the CPPO-specific symbols

| Notations | Corresponding Meanings |
| --- | --- |
| $i(1, ..., I)$ | generation of the i-th token |
| $t(1, ..., T)$ | t-th task of CL |
| $x$ | a rollout sample |
| $x_i$ | i-th token of sample $x$ |
| $s_i$ | state-i: prompt + $x_{1:i-1}$ |
| $\theta$ | parameters of policy learned in task-t |
| $\pi_t$ | policy learned in task-t |
| $\mathbf{P}_{\pi_t(\mathbf{x})}$ | generation probability of $x$ under $\pi_t$ |
| $\mathbf{J}(\theta)$ | total objective of PPO or CPPO |
| $L^{CLIP}$ | clipped policy learning objective |
| $L^{VF}$ | squared-error value loss |
| $V(s_i)$ | value estimation by critic |
| $\lambda / \gamma$ | reward / value discount coefficients |
| $\mathbf{A}_i$ | advantage score of token $x_i$ |
| $\mathbf{R}(x)$ | reward model score of $x$ |
| $r_i(\theta)$ | the probability ratio |
| $\epsilon$ | clip hyperparameter |
| $clip(\cdot, 1 \pm \epsilon)$ | clip by $1 \pm \epsilon$ |
| $c$ | coefficient of $L^{VF}$ |
| $N$ | samples number per rollout batch |
| $k$ | *the threhold of times of standard variance* |
| $L^{KR}$ | *knowledge retention penalty* |
| $\alpha(x)$ | *weight of policy learning* |
| $\beta(x)$ | *weight of knowledge retention* |
| $ub, lb$ | *the upper bound and lower bound of weights* |
| $\mu[\mathbf{P}_{\pi_\theta}]$ | *expectation of $\mathbf{P}_{\pi_\theta}(x)$* |
| $\mu[\mathbf{R}]$ | *expectation of $\mathbf{R}(x)$* |
| $\sigma[\mathbf{P}_{\pi_\theta}]$ | *standard variance of $\mathbf{P}_{\pi_\theta}(x)$* |
| $\sigma[\mathbf{R}]$ | *standard variance of $\mathbf{R}(x)$* |
| $\phi \, (|\phi|=2N)$ | *parameters for weight learning* |
| $\mathbf{L}_{coef}(\phi)$ | *objective of weight learning* |

# B THE THEORETICAL ANALYSIS OF CPPO

The theoretical objective in Eq. 4 is an intuitive implementation of our basic idea. Based on it, we derive a more practical objective in Eq. 6. Next, we will elaborate the relationship between the two and explain how we were inspired by Eq. 4 and designed Eq. 6.

**Eq. 6 is a generalized version of Eq. 4.**

Let $I_{D_1}(x)$ and $I_{D_2}(x)$ denote the indicator functions of the sets of $D_1$ and $D_2$, respectively. In Eq. 6, $\alpha(x)$ and $\beta(x)$ can be any non-negative real-valued functions defined on the rollout set. We claim that in Eq. 4, $\alpha(x)$ and $\beta(x)$ are specialized as $\alpha(x) = I_{D_1}(x)$, $\beta(x) = I_{D_2}(x)$. Next, we provide the derivation step by step.

**By introducing the actor-critic version, the clipped ratio, the objective Eq. 4. can be written as the objective Eq. 5.**

1. We utilize notations $I_{D_1}(x)$ and $I_{D_2}(x)$ to rewrite the Eq.4 as $\max_\theta \mathbb{E}_{x \sim \pi_\theta} I_{D_1}(x) \cdot \mathbf{R}(x) - \mathbb{E}_{x \sim \pi_{t-1}} I_{D_2}(x) \cdot D_{KL}(\mathbf{P}_{\pi_\theta}(x) \parallel \mathbf{P}_{\pi_{t-1}}(x))$.

2. Then we introduce the importance sampling like PPO, the above objective can be written as $\max_\theta \mathbb{E}_{x \sim \pi_{t-1}} I_{D_1}(x) \cdot \frac{\mathbf{P}_{\pi_\theta}(x)}{\mathbf{P}_{\pi_{t-1}}(x)} \mathbf{R}(x) - \mathbb{E}_{x \sim \pi_{t-1}} I_{D_2}(x) \cdot D_{KL}(\mathbf{P}_{\pi_\theta}(x) \parallel \mathbf{P}_{\pi_{t-1}}(x))$.

3. In the PPO method, the objective is to maximize the expectation of the advantage function instead of the reward value. By introducing the advantage function instead of the reward, the above objective can be written as $\max_\theta \mathbb{E}_{x \sim \pi_{t-1}} I_{D_1}(x) \cdot \frac{\mathbf{P}_{\pi_\theta}(x)}{\mathbf{P}_{\pi_{t-1}}(x)} \mathbf{A}(x) - \mathbb{E}_{x \sim \pi_{t-1}} I_{D_2}(x) \cdot D_{KL}(\mathbf{P}_{\pi_\theta}(x) \parallel \mathbf{P}_{\pi_{t-1}}(x))$.

4. By introducing the knowledge retention penalty, the above objective is written as:
$\max_\theta \mathbb{E}_{x \sim \pi_{t-1}} I_{D_1}(x) \cdot \frac{\mathbf{P}_{\pi_\theta}(x)}{\mathbf{P}_{\pi_{t-1}}(x)} \mathbf{A}(x) - \mathbb{E}_{x \sim \pi_{t-1}} I_{D_2}(x) \cdot L^{KR}(x).$

5. In the CL task, the new policy $\pi_t$ is generally initialized by the old policy $\pi_{t-1}$. In CPPO, we treat the $\pi_{t-1}$ and $\pi_t$ as the reference model and policy model respectively. Then, we consider the actor-critic version, the clipped ratio, and the entropy bonus used in PPO[32], the above objective can be written as $\mathbf{J}(\theta)^{'} = L^{\boldsymbol{I_{D_1}} \cdot CLIP + \boldsymbol{I_{D_2}} \cdot KR + VF}(\theta).$

**From the objective Eq. 5 to the objective Eq. 6.**

Comparing $\mathbf{J}(\theta)^{'}$ with $\mathbf{J}(\theta) = L^{\boldsymbol{\alpha} \cdot CLIP + \boldsymbol{\beta} \cdot KR + VF}(\theta)$ in Eq. 6, $\alpha(x)$ and $\beta(x)$ are specialized as $I_{D_1}(x)$ and $I_{D_2}(x)$. Hence, the objective Eq. 5 can be treated as a special case of Eq. 6.

## C  CASE STUDY OF SUMMARY TASK

Table 10 presents a case where CPPO improves the performance on task-1 (r/relationships) after learning task-2 (r/others).

Table 10: Example responses of Reddit TL;DR.

| |
|---|
| **The original POST** (the domain is r/relationships): |
| TITLE: [20F] My [22M] SO never wants to spend time with me. |
| POST: My boyfriend and I have been together off & on for three years. We've had our fair share of issues, but I love him and I think he loves me. |
| We work opposite shifts, and it really cuts into our time together. Majorly. On the few days we both have off, every other weekend, he chooses to hang out with his friends. He works with his friends on the same schedule, and they are over every single day. |
| I always feel like I have to force him to do anything with me and I'm getting tired of it. I always complain to him that it would be nice if we could spend time together. But, he just says in over reacting. And that I can hang out with him and his friends. The thing is, we haven't spent a day together just him and I in over a year. I like his friends but I just want a day where we can be together just us. I'm sick of being ignored for his friends, or video games and I don't know what to do. |
| **TL;DR:** |
| **The summary after learning task-1 (r/relationships)**: |
| **CPPO (rPMS: 1.75):** My SO never wants to spend time with me. He works with his friends on the same schedule, and they are over every single day. I always feel like I have to force him to do anything with me and I'm getting tired of it. |
| **The summary after learning task-2 (r/others)**: |
| **CPPO (rPMS: 1.82):** My boyfriend prioritizes friends over time with me. Opposite work schedules strain time together. I'm tired of urging for attention. Seeking advice for reconnection. |
| **PPO(In order) (rPMS: 1.44):** The boyfriend has been with his boyfriend for three years and doesn't want to spend time with his boyfriend anymore. The boyfriend's boyfriend has been with him for three years and doesn't want to spend time with him anymore. |
| **PPO (rPMS:1.65):** The boyfriend and I have been dating for three years and I love him but he is not interested in going out with his girlfriend. |

After learning the task-2, the model generates an accurate and concise summary. It more clearly conveys the main issue and emotions in the post, along with the desire for advice. And it also gets a higher rPMS (1.82 v.s. 1.75).

From the three summaries after learning task-2, it can be observed that PPO(In order) seems to exhibit a more noticeable knowledge forgetting, with a seeming lack of understanding of the concept "boyfriend." This is due to the frequent occurrence of "boyfriend" in task-1 (r/relationships) and its almost absence in task-2 (r/others), resulting in a case of catastrophic forgetting. The PPO model still manages to convey the main essence of the text, but it overlooks some crucial details, such as "opposite work schedule" and "prioritizes friends over time with me", hence PPO lags behind CPPO in terms of rPMS value.

# D  BASELINES

**Supervise fine-tuning (SFT)** directly learns the human-labeled summary through the cross-entropy loss.

**Online L2Reg** penalizes the updating of model parameters through a L2 loss $L_2^t(\theta) = \sum_i (\theta_t^i - \theta_{t-1}^i)^2$. This regularization term mitigates the forgetting issue by applying a penalty for every parameter change.

**EWC** (Kirkpatrick et al., 2017) uses fisher information to measure the parameter importance to old tasks, then slows down the update of the important parameters by L2 regularization.

**MAS** (Aljundi et al., 2018) computes the importance of the parameters of a neural network in an unsupervised and online manner to restrict the updating of parameters in the next task.

**LwF** (Li & Hoiem, 2018) is a knowledge-distillation-based method, which computes a smoothed version of the current responses for the new examples at the beginning of each task, minimizing their drift during training.

**TFCL** (Aljundi et al., 2019) proposes to timely update the importance weights of the parameter regularization by detecting plateaus in the loss surface.

**PC** (Kaplanis et al., 2019) is inspired by the biologically plausible synaptic model and proposes to consolidate memory directly at the behavioral level by knowledge distillation, aiming to mitigate catastrophic forgetting in the reinforcement learning context.

**HN-PPO** (Schöpf et al., 2022) Hypernetwork-PPO is a continual model-free RL method employing a hyper network to learn multiple policies in a continual manner by using PPO.

**NLPO** (Ramamurthy et al., 2022) NLPO learns to mask out less relevant tokens in-context as it trains via top-p sampling, which restricts tokens to the smallest possible set whose cumulative probability is greater than the probability parameter $p$ (Holtzman et al., 2018).

# E  DISCUSSION

## E.1  HYPERPARAMETER SENSITIVE ANALYSIS

Due to the introduction of additional hyperparameters by CPPO, we conducted a sensitivity analysis of CPPO's hyperparameters. We conduct sensitivity analysis on five hyperparameters, including the threhold of times of standard variance $k$, the upper bound $ub$ and lower bound $lb$ of weights, the learning rate **weights-lr** of CPPO Heuristic, and the multiplier $\tau$. As shown in Table 11, the analysis of experimental results shows that our method is insensitive to the introduction of extra hyperparameters.

Table 11: Hyperparameter sensitivity analysis of CPPO Heuristic and CPPO Learn.

| Hyper-Parameters k / ub / lb | Method | Task-1 $\text{rPMS}_1$ ($\uparrow$) | rouge ($\uparrow$) | AT ($\downarrow$) | Task-2 $\text{rPMS}_2$ ($\uparrow$) | rouge ($\uparrow$) | SFR ($\downarrow$) |
|---|---|---|---|---|---|---|---|
| 0.85 / 2.5 / 0.5 | Heuristic | **3.020** ±0.137 | 0.213 ±0.024 | 0.035 ±0.023 | **2.978** ±0.113 | 0.174 ±0.019 | -0.164 ±0.009 |
| k: 0.85 -> 0.5 | Heuristic | 3.011 ±0.141 | 0.209 ±0.026 | 0.036 ±0.025 | 2.97 ±0.121 | 0.171 ±0.018 | -0.162 ±0.011 |
| k: 0.85 -> 1.0 | Heuristic | 3.017 ±0.132 | 0.214 ±0.025 | 0.031 ±0.031 | 2.891 ±0.117 | 0.170 ±0.021 | -0.151 ±0.010 |
| ub: 2.5 -> 1.5 | Heuristic | 2.982 ±0.124 | 0.205 ±0.031 | 0.040 ±0.042 | 2.809 ±0.124 | 0.173 ±0.023 | -0.165 ±0.008 |
| ub: 2.5 -> 3.0 | Heuristic | 3.012 ±0.123 | 0.205 ±0.042 | 0.040 ±0.051 | 2.941 ±0.115 | 0.171 ±0.029 | **-0.166** ±0.013 |
| lb: 0.5 -> 0.1 | Heuristic | 3.011 ±0.162 | **0.221** ±0.051 | **0.024** ±0.041 | 2.809 ±0.115 | 0.167 ±0.019 | -0.162 ±0.011 |
| lb: 0.5 -> 0.0 | Heuristic | 2.997 ±0.152 | 0.219 ±0.031 | 0.026 ±0.040 | 2.941 ±0.141 | **0.179** ±0.016 | -0.161 ±0.010 |
| **weights-lr / $\tau$** | **Method** | $\text{rPMS}_1$ ($\uparrow$) | **rouge** ($\uparrow$) | **AT** ($\downarrow$) | $\text{rPMS}_2$ ($\uparrow$) | **rouge** ($\uparrow$) | **SFR** ($\downarrow$) |
| 0.01 / 0.1 | Learn | **3.180** ±0.154 | **0.220** ±0.040 | **0.028** ±0.042 | **3.085** ±0.134 | 0.164 ±0.024 | -0.161 ±0.008 |
| weights-lr: 0.01 -> 0.1 | Learn | 3.122 ±0.162 | 0.201 ±0.041 | 0.044 ±0.041 | 2.824 ±0.141 | 0.171 ±0.024 | -0.155 ±0.012 |
| weights-lr: 0.01 -> 0.5 | Learn | 3.141 ±0.131 | 0.209 ±0.053 | 0.036 ±0.052 | 2.934 ±0.125 | 0.170 ±0.022 | -0.162 ±0.016 |
| $\tau$: 0.1 -> 0.01 | Learn | 3.042 ±0.137 | 0.211 ±0.046 | 0.034 ±0.054 | 2.89 ±0.151 | 0.168 ±0.026 | -0.161 ±0.020 |
| $\tau$: 0.1 -> 0.5 | Learn | 3.087 ±0.151 | 0.212 ±0.040 | 0.033 ±0.062 | 2.892 ±0.118 | **0.174** ±0.032 | -0.161 ±0.014 |
| $\tau$: 0.1 -> 1.0 | Learn | 3.072 ±0.148 | 0.209 ±0.039 | 0.036 ±0.051 | 2.967 ±0.129 | 0.172 ±0.021 | **-0.169** ±0.008 |

## E.2 COMPLEXITY ANALYSIS

In this section, we compare CPPO (learnable weights) with PPO in terms of time and memory occupation. The steps of CPPO are similar to PPO, except for the step of learning balance weights. By considering the time of the rollout step as our reference, we demonstrate that the time required to learn the weights is negligible compared to the overall training process of CPPO and PPO. Figure 5 illustrates the time required for learning balance weights and the time for making rollouts during the training of gpt2-s and gpt2-xl. For gpt2-s training, the ratio between the time spent on learning balance weights (approximately 8s) and the time taken for rollout steps (around 400s) is 1/50. This ratio decreases to 1/200 when training gpt2-xl, due to the fact that the time for learning balance weights remains the same, while the time for making rollouts increases to 1600s. Hence, our method does not significantly increase the time complexity of PPO, especially for training large language models.

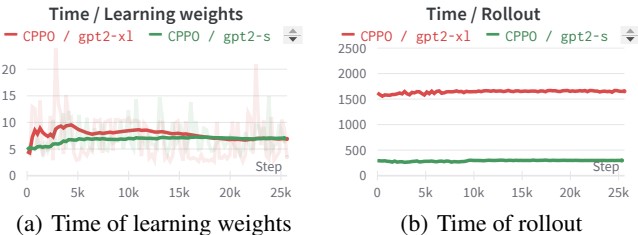

(a) Time of learning weights       (b) Time of rollout

Figure 5: Time of learning weights and time of making rollout.

For memory occupation, we record the GPU memory allocation, GPU utilization, and the process memory in the training process of PPO and CPPO. Figure 6 illustrates the comparison of the above metrics between PPO and CPPO. CPPO, which learns the balance weights and calculates the knowledge retention loss, leads to higher allocation of GPU memory and process memory compared to PPO. Nevertheless, the improvements in GPU memory and process memory are not particularly substantial.

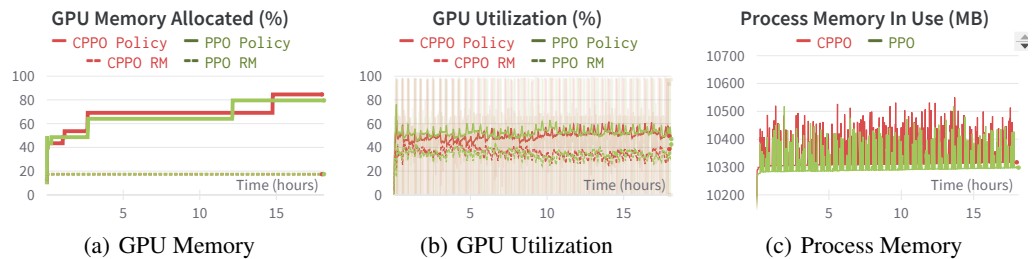

(a) GPU Memory       (b) GPU Utilization       (c) Process Memory

Figure 6: GPU utilization and memory allocation when the algorithm runs for 15+ hours. Compared to the PPO method, our CPPO does not significantly utilize extra memory.

## F TASKS FOR STATIC LEARNING

We compare PPO and CPPO on 3 static learning tasks, including random walks, sentiment text generation, and summary on CNN Daily Mail.

### F.1 RANDOM WALKS

The task(Chen et al., 2021) involves finding the shortest path on a directed graph. The reward is based on how optimal the path is compared to the shortest possible (bounded in [0, 1]). Paths are represented as strings of letters, with each letter corresponding to a node in the graph. For CPPO or

PPO, a language model was fine-tuned to predict the next token in a sequence of returns-to-go (sum of future rewards), states, and actions.

## F.2 SENTIMENT TEXT GENERATION

This task focuses on generating positive movie reviews by fine-tuning a pre-trained model on the IMDB dataset using a sentiment reward function. We consider the IMDB(Maas et al., 2011) dataset for the task of generating text with positive sentiment. The dataset consists of 25k training, 5k validation and 5k test examples of movie review text with sentiment labels of positive and negative. We utilize a sentiment classifier (Sanh et al., 2019) trained on pairs of text and labels as a reward model, which provides sentiment scores indicating how positive a given piece of text is.

## F.3 SUMMARY ON CNN DAILY MAIL

The dataset for this task comprises 287k training examples, 13k validation examples, and 11k test examples. We utilize meteor(Banerjee & Lavie, 2005) as the reward function. T5 is chosen as the base language model due to its pre-training in a unified text-to-text framework and its ability to handle zero-shot capabilities.

## F.4 EVALUATION ON NON-CONTINUAL LEARNING TASKS

We compare the performance of PPO and CPPO on three static learning tasks, including randomwalks(Chen et al., 2021), sentiment text generation (Ramamurthy et al., 2022) on IMDB(Huang et al., 2021), and summarization on CNN Daily Mail (Hermann et al., 2015). As in the continual learning setting, we initialize our model with a pre-trained model and compute the knowledge retention penalty using both the policy model and the pre-trained model. Experimental results demonstrate that CPPO outperforms PPO in static learning settings. We observe the instability of PPO on the sentiment text generation task, while CPPO can learn stably. As shown in Figure 7, CPPO outperforms the PPO algorithm on all three tasks, which is attributed to CPPO's ability to enhance the learning of high-performance, high-variance, and overfitting samples while slowing down the learning of noisy samples.

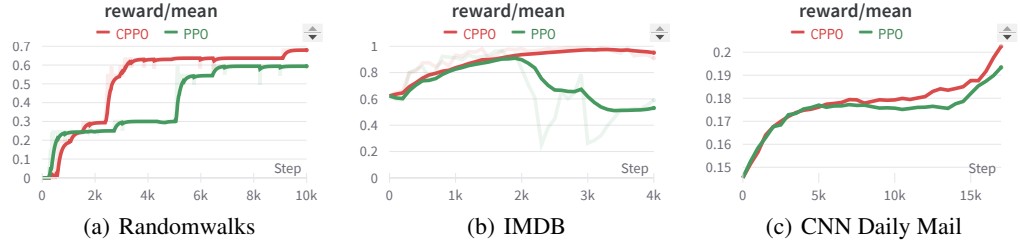

(a) Randomwalks       (b) IMDB       (c) CNN Daily Mail

Figure 7: Evaluation results on the test data during different training steps. a) The optimality scores in [0, 1], as compared to the shortest path. b) Positive sentiment scores provided by the distilbert trained on the IMDB dataset. c) METEOR (Metric for Evaluation of Translation with Explicit ORdering).

# G DETAILS OF IMPLEMENTATION

The algorithm of CPPO is presented in Algorithm 1. Step 3 is for learning an RM continually; step 7 is for computing balance weights; step 9 is for calculating CPPO loss; other steps are the same as the PPO algorithm. Our implementation is based on the open source library trlx[6]. The experiments on the SHP dataset are conducted in 4 Nvidia A100 GPUs with 80 GB of RAM, other experiments are conducted in 2 Nvidia Tesla V100 GPUs with 32 GB of RAM. To conserve GPU memory, we utilize CPU-Offload and Mixed-Precision techniques. We provide all hyperparameters used in both the PPO and CPPO algorithms in Table 12.

---

[6]https://github.com/CarperAI/trlx

---

**Algorithm 1:** CPPO

---

**input** : SFT model $M_{SFT}$, critic model $C$, reward model $RM$, ppo epoches $N$, ppo steps $S$, query streams $\mathbf{Q_t}(t = 1, 2, ..., T)$.

**output** : Aligend model $M_T^{'}$.

1 Initialize actor $M_0 \leftarrow M_{SFT}$ ;
2 **for** *t = 1,2,...,T* **do**
3      update $RM$ on new feedback by MAS;
4      **for** *epoch = 1,2,...,N* **do**
5          make actor $M_{t-1}$ generate response $O_{t-1}$ on prompts $Q_t$ ;
6          compute generation probability $P_{\pi_{t-1}}(x)$ of $M_{t-1}$ on $O_{t-1}$, reward $R(x)$ of response $O_{t-1}$ by $RM$, state value evaluation $v_{t-1}$ by critic $C$ and advantage $A_{t-1}(x)$;
7          compute a set of balance weights $\{(\alpha(x), \beta(x))|x \in O_{t-1}\}$ ;
8          **for** *step = 1,2,...,S* **do**
9              compute the CPPO loss by Equation (5) ;
10              update model $M_t$ by Adam optimizer ;
11          **end**
12      **end**
13 **end**

---

Table 12: Hyperparameters of different tasks. *Italic* font denotes the CPPO-specific hyperparameters. For all tasks, we utilize the default PPO hyperparameters released by trlx.

| | CNN | Random walks | IMDB | Reddit |
|---|---|---|---|---|
| **seq-length** | 612 | 10 | 1024 | 550 |
| **total-steps** | 17200 | 10000 | 4000 | 25600 |
| **batch-size** | 12 | 100 | 128 | 8 |
| **model (huggingface)** | google/flan-t5-small | CarperAI/randomwalks | lvwerra/gpt2-imdb | gpt2 |
| **num-layers-unfrozen** | 2 | -1 | 2 | 8 |
| **optimizer** | adamw | adamw | adamw | adamw |
| **lr** | 1.00E-05 | 3.00E-04 | 1.00E-04 | 5.00E-06 |
| **betas** | [0.9, 0.999] | [0.9, 0.95] | [0.9, 0.95] | [0.9, 0.999] |
| **eps** | 1.00E-08 | 1.00E-08 | 1.00E-08 | 1.00E-08 |
| **weight-decay** | 1.00E-06 | 1.00E-06 | 1.00E-06 | 1.00E-06 |
| **lr scheduler** | cosine-annealing | cosine-annealing | cosine-annealing | cosine-annealing |
| **T-max** | 17200 | 10000 | 4000 | 25600 |
| **eta-min** | 1.00E-06 | 3.00E-04 | 1.00E-04 | 5.00E-06 |
| **num-rollouts** | 512 | 128 | 128 | 512 |
| **chunk-size** | 12 | 128 | 128 | 32 |
| **ppo-epochs** | 4 | 4 | 4 | 4 |
| **init-kl-coef** | 0.05 | 0.05 | 0.05 | 0.1 |
| **target** | 6 | 6 | 6 | 6 |
| **horizon** | 10000 | 10000 | 10000 | 10000 |
| **gamma** | 0.99 | 1 | 1 | 1 |
| **lam** | 0.95 | 0.95 | 0.95 | 0.95 |
| **cliprange** | 0.2 | 0.2 | 0.2 | 0.2 |
| **cliprange-value** | 0.2 | 0.2 | 0.2 | 0.2 |
| **vf-coef** | 1 | 1.2 | 1 | 0.2 |
| **scale-reward** | False | False | False | False |
| **cliprange-reward** | 10 | 1 | 10 | 10 |
| **max-new-tokens** | 100 | 9 | 40 | 50 |
| **top-k** | 50 | - | - | - |
| **top-p** | 0.95 | - | - | - |
| *k* | 0.85 | 0.85 | 0.85 | 0.85 |
| *reg-coef* | 0.1 | 0.1 | 0.1 | 0.1 |
| *ub* | 2.5 | 2.5 | 2.5 | 2.5 |
| *lb* | 0.5 | 0.2 | 0.5 | 0.5 |
| *weights-lr* | 0.01 | 0.01 | 0.01 | 0.01 |

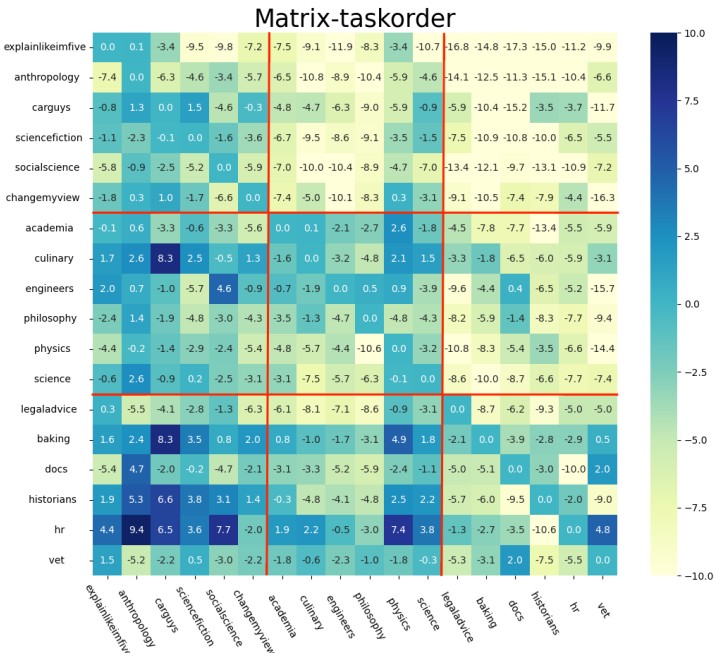

Figure 8: Cross-domain evaluation of 18 reward models. The i-th row represents the RM trained on domain i, the j-th column represents the RM tested on domain j, and the value indicates the accuracy degradation value, namely, $Acc_{test}^j - Acc_{test}^i$.

## H    EVALUATION OF STANFORD HUMAN PREFERENCES BENCHMARK

We have established a novel continual RLHF benchmark using the Stanford Human Preferences Dataset (SHP) (Ethayarajh et al., 2022). We assess CPPO and baselines on this benchmark within the context of a 3-task sequence setting.

### H.1    DOMAIN INCREMENTAL LEARNING SETTING

The policy is required to continuously learn from three segments of the **SHP** dataset. The SHP dataset comprises 18 domains, which we divide into 3 parts based on the maximal errors of the out-of-distribution (OOD) generalization(Shen et al., 2021). We employ the *SteamSHP-flan-t5-xl model* (Ethayarajh et al., 2022), developed by Stanford, as the reference preference model (rPM) for assessing responses to SHP prompts. The *SteamSHP-flan-t5-xl model*[7] is trained on the combination of the SHP (all domains) and the HH-RLHF(Bai et al., 2022a) human preference data.

To create more challenging DIL tasks, we individually trained 18 reward models (based on Llama-7b) on 18 domains of SHP data and evaluated each reward model on the test set of the 18 domains, resulting in an accuracy difference matrix of size 18x18 as shown in Figure 8. In this matrix, the row coordinates represent the training domains, and the column coordinates represent the evaluation domains. The elements in the matrix represent the relative decrease in performance on the evaluation domain compared to the training domain (both evaluated on test sets from various domains). Based on this accuracy difference matrix, we divided the 18 domains into 3 groups (each has 6 domains). This division ensures that there will be a significant performance decrease, i.e., the largest error of OOD generalization, when evaluated on domains from different groups. For example, the RM trained in the first group (from "explainlikeimfive" to "changemyview") has large errors in the third group (from "legaladivce" to "vet"). Refer to Table 13 for the distribution of samples across various domains.

---

[7]https://huggingface.co/stanfordnlp/SteamSHP-flan-t5-xl

Table 13: Number of posts in the SHP dataset by subreddit.

| Task ID | subreddit | train | valid | test | total | % of All |
|---|---|---|---|---|---|---|
| | ALL | 348718 | 18436 | 18409 | 385563 | 100.00% |
| Task 1 | explainlikeimfive | 19592 | 1014 | 1070 | 21676 | 5.62% |
| | askanthropology | 3910 | 203 | 268 | 4381 | 1.14% |
| | askcarguys | 3227 | 159 | 117 | 3503 | 0.91% |
| | asksciencefiction | 29382 | 1576 | 1987 | 32945 | 8.54% |
| | asksocialscience | 2706 | 147 | 188 | 3041 | 0.79% |
| | changemyview | 38173 | 1637 | 1836 | 41646 | 10.80% |
| Task 2 | askacademia | 31450 | 2095 | 1708 | 35253 | 9.14% |
| | askculinary | 45710 | 2094 | 2563 | 50367 | 13.06% |
| | askengineers | 57096 | 3154 | 2638 | 62888 | 16.31% |
| | askphilosophy | 10307 | 608 | 677 | 11592 | 3.01% |
| | askphysics | 7364 | 409 | 587 | 8360 | 2.17% |
| | askscience | 13316 | 899 | 977 | 15192 | 3.94% |
| Task 3 | legaladvice | 21170 | 1106 | 1011 | 23287 | 6.04% |
| | askbaking | 44007 | 2096 | 1544 | 47647 | 12.36% |
| | askdocs | 6449 | 315 | 455 | 7219 | 1.87% |
| | askhistorians | 3264 | 113 | 164 | 3541 | 0.92% |
| | askhr | 8295 | 641 | 395 | 9331 | 2.42% |
| | askvet | 3300 | 170 | 224 | 3694 | 0.96% |

## H.2 EVALUATION METRIC FOR CONTINUAL LEARNING

**Overall performance** is commonly assessed using *average accuracy* (AA)(Chaudhry et al., 2018; Lopez-Paz & Ranzato, 2017) and *average incremental accuracy* (AIA) (Douillard et al., 2020; Hou et al., 2019). In our evaluation framework, accuracy is replaced by the reference Preference Score, which ranges from 0 to 1. Let $a_{k,j} \in [0,1]$ represent the reference Preference Score evaluated on the test set of the $j$-th task after incremental learning of the $k$-th task ($j \leq k$). The two metrics at the $k$-th task are then defined as:

$$\text{AA}_k = \frac{1}{k} \sum_{j=1}^{k} a_{k,j}, \tag{8}$$

$$\text{AIA}_k = \frac{1}{k} \sum_{i=1}^{k} \text{AA}_i, \tag{9}$$

where AA signifies the current overall performance, and AIA additionally captures the historical fluctuations.

**Memory stability** can be evaluated through *forgetting measure* (FM)(Chaudhry et al., 2018) and *backward transfer* (BWT) (Lopez-Paz & Ranzato, 2017). Regarding the former, the forgetting of a task is computed as the difference between its best performance achieved in the past and its current performance:

$$f_{j,k} = \max_{i \in \{1,\dots,k-1\}} (a_{i,j} - a_{k,j}), \forall j < k. \tag{10}$$

FM at the $k$-th task is the average forgetting of all old tasks:

$$\text{FM}_k = \frac{1}{k-1} \sum_{j=1}^{k-1} f_{j,k}. \tag{11}$$

Regarding the latter, BWT assesses the collective impact of learning the $k$-th task on all previous tasks:

$$\text{BWT}_k = \frac{1}{k-1} \sum_{j=1}^{k-1} (a_{k,j} - a_{j,j}), \tag{12}$$

## H.3 EVALUATION RESULTS

We utilize the Llama-7B as the backbone of the policy model and reward model. When training the reward model, we fine-tune the all of parameters. When training the RLHF model, we only fine-tune the top 8 layers.

The experiments are run for 3 random seeds, the final results are shown in Table 14. We found that using reward and policy models based on Llama-7B, as opposed to training with gpt2-xl as the backbone, significantly improves the training stability. It is related to larger reward models being less susceptible to reward attacks(Gao et al., 2022).

From Table 14, we observe that the Iterated RLHF method has a significant effect in preventing forgetting. Since it re-trains the reward model and policy model from scratch at each task, resulting in higher memory and computational complexity. CPPO, compared to PPO, shows a noticeable improvement in overall performance, and it outperforms PPO in the Memory stability metric. This indicates that CPPO can effectively learning varying human preferences.

Table 14: Performance on DIL setting. The range of SteamSHP score is $[0, 1]$.

| Method | Task 1 SteamSHP (↑) | Task 2 SteamSHP (↑) | Task 3 SteamSHP (↑) | Overall performance AA (↑) | Overall performance AIA (↑) | Memory stability BWT (↑) | Memory stability FM (↓) |
|---|---|---|---|---|---|---|---|
| SFT (In order) | 0.806 ±0.0101 | 0.836 ±0.0103 | 0.853 ±0.0103 | 0.832 ±0.0061 | 0.837 ±0.0039 | -0.022 ±0.0094 | 0.022 ±0.0094 |
| SFT (multi-tasks) | 0.831 ±0.0266 | 0.847 ±0.0145 | 0.858 ±0.0114 | 0.845 ±0.0147 | 0.844 ±0.0082 | -0.006 ±0.0183 | 0.009±0.0160 |
| Iterated RLHF | 0.869 ±0.0583 | 0.88 ±0.0490 | 0.887 ±0.0421 | 0.879 ±0.0488 | 0.874 ±0.0433 | **-0.0004** ±0.0186 | **0.003** ±0.0162 |
| PPO | 0.853 ±0.0573 | 0.879 ±0.0428 | 0.889 ±0.0369 | 0.874 ±0.0433 | 0.877 ±0.0378 | -0.017 ±0.0351 | 0.020 ±0.0327 |
| CPPO (Heuristic) | 0.864 ±0.0557 | 0.89 ±0.0448 | 0.894 ±0.0350 | 0.883 ±0.0429 | 0.887 ±0.0391 | -0.015 ±0.0321 | 0.018 ±0.0300 |
| CPPO (Learn) | **0.872** ±0.0544 | **0.898** ±0.0450 | **0.899** ±0.0342 | **0.89** ±0.0424 | **0.894** ±0.0389 | -0.013 ±0.0298 | 0.016 ±0.0281 |

# I   LIMITATION: THE RISK OF OVER-OPTIMIZATION

We have observed that both PPO and CPPO have the potential risk of achieving high rewards while generating poor summaries. This issue is depicted in Figure 9, where the policy model tends to overoptimize against the RM when trained for 100k steps (390 epochs). Over time, the policy becomes excessively focused on maximizing rewards without adequately considering the quality of the generated summaries. To address the risk of optimization, various strategies can be employed. One approach is to train an additional RM to evaluate the policy during training. This allows for evaluating the policy's performance using an external objective metric, providing a more robust measure of the summary quality. Another strategy is to implement early stopping, where the training process is halted based on the quality of the generated summaries or other external metrics. Instead of solely focusing on maximizing rewards, we prioritize the quality of the generated summaries. Training is halted when the summary quality reaches a certain threshold or shows no further improvement. This approach ensures that the generated summaries not only maximize rewards but also maintain a high level of quality.

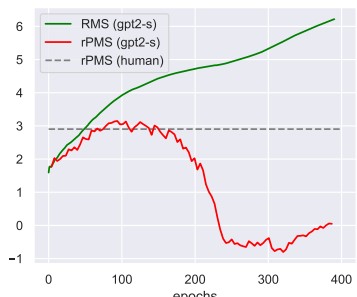

Figure 9: Overoptimize against the reward model. After training 100k steps, the RM score (RMS) on the test data has a high bias compared with the rPMS.

Recent research (Gao et al., 2022) has noted an interesting observation regarding larger policy models. It has been found that as the size of the policy models increases, they become less susceptible to over-optimization against the RM. This suggests that scaling up the model size can potentially alleviate the over-optimization issue by introducing more complexity and capacity into the policy model, making it harder for the model to excessively optimize solely for rewards without considering the summary quality.

In summary, mitigating the risk of over-optimization in PPO and CPPO can be achieved through strategies such as training additional reward models, implementing early stopping, and considering larger policy models. These measures aim to strike a balance between achieving high rewards and generating high-quality summaries, ensuring that the models generalize well and produce reliable results even on unseen data.

## J  BROADER IMPACT

The broader impact of our proposed CPPO method is significant for both researchers and practitioners in the field of NLP. By addressing the limitations of existing RLHF-based LMs, we enable the continual alignment of these models with human preferences, opening up new possibilities for their widespread adoption and deployment.

One important implication of our work is the reduction of time and computational costs associated with retraining LMs. In many real-world scenarios, complete retraining is impractical due to resource constraints and data privacy. By introducing sample-wise weights and enhancing policy learning while retaining valuable past experiences, CPPO offers a more efficient and practical alternative. This efficiency allows practitioners to keep LMs up-to-date with evolving human preferences without incurring the substantial overhead of retraining, making them more accessible and applicable across a range of applications.

The practical implications of our work extend beyond research and development. Industries that heavily rely on LMs, such as customer service, virtual assistants, and content generation, stand to benefit from the continual alignment provided by CPPO. The improved performance and adaptability of LMs enable more personalized and effective interactions with users, enhancing user satisfaction and overall user experience. Additionally, CPPO's ability to align with human preferences consistently enables the development of more inclusive and fair AI systems that better understand and respect diverse user needs and values.

In summary, our CPPO method has broad implications for the NLP community and beyond. By addressing the challenges associated with RLHF-based LMs, our approach offers a practical and efficient solution for continually aligning with human preferences while reducing retraining costs and preserving data privacy. These advancements promote the wider adoption and responsible use of LMs in various domains, leading to more personalized, inclusive, and trustworthy AI systems.

