# OpenReview forum: "CPPO: Continual Learning for Reinforcement Learning with Human Feedback"
_ICLR.cc/2024/Conference — ICLR 2024 poster_

### Official Review · Reviewer_CUAu · 2023-10-19

**Soundness:** 3 good
**Presentation:** 2 fair
**Contribution:** 3 good
**Rating:** 6
**Confidence:** 4

**Summary:**

This paper studies the problem in the continual learning problem in the RLHF module of training language models. The authors propose continual PPO (CPPO), which utilizes a sample-wise weighting strategy based on the vanilla PPO algorithm to balance policy learning and knowledge retention. Experimental results show that CPPO has a good continual learning ability in the domain incremental learning summary task.

**Strengths:**

- This paper studies the important problem of reinforcement learning with human feedback (RLHF) in a continual manner due to the limitation of computational cost or data privacy, which may have a great impact to the LLM community.
- The paper is well written and easy to follow.
- CPPO outperforms multiple baselines in the proposed summarization benchmark.

**Weaknesses:**

- The major contribution of this paper is to propose the problem of RLHF in a continual manner, however, the motivation and necessity of this problem is lack of detailed explanation and support, hindering readers’ understanding of its value. The authors should introduce more related work that retrain or finetune the LM when new data arrive. Furthermore, the storage and computational cost of different approaches should be analyzed, for example, the retraining methods [1], replay-based methods [2] and regularization-based methods [3].
- The experiment part is weak.
    - In the main experiment to evaluate the continual learning ability of CPPO in NLP tasks, only one benchmark Reddit TL;DR summarization is considered. What’s more, the length of the continual task stream is only 2, which is not enough for a solid evaluation of continual learning ability compared to related work [4][5][6][7].
    - The proposed CPPO method can also be applied to other continual reinforcement learning tasks other than RLHF. Simpler conventional control tasks like DoorGym [4][5] should be included to further validate the soundness of CPPO.
- The mathematical symbols and spellings need to be checked, for example,
    - $A_i$ in Eq.(3) was not mentioned before.
    - explanation of Eq.(4), k is a hyperparameter → $k$ is a hyperparameter.
    - In Eq.(6), coefficients should be $C_1, C_2$, not $r_1, r_2$.
    - Experiment settings: lkie → like.

Reference:

[1] Yuntao Bai, et al. Training a helpful and harmless assistant with reinforcement learning from human feedback. arxiv 2022.

[2] David Rolnick, et al. Experience Replay for Continual Learning. NeurIPS 2019.

[3] James Kirkpatrick, et al. Overcoming catastrophic forgetting in neural networks. PNAS, 2017.

[4] Yizhou Huang, et al. Continual model-based reinforcement learning with hypernetworks. ICRA 2021.

[5] Philemon Schöpf, et al. Hypernetwork-PPO for continual reinforcement learning. NeurIPS 2022.

[6] Samuel Kessler, et al. Same state, different task: Continual reinforcement learning without interference. AAAI 2022.

[7] Anonymous, et al. Scalable Language Model with Generalized Continual Learning. Summited to ICLR 2024. https://openreview.net/forum?id=mz8owj4DXu.

**Questions:**

see weeknesses

---

> ### Author Response · Authors · 2023-11-20
> **Response to Reviewer CUAu**
>
> Thank you for your helpful comments! We appreciate your comments that our method may have a great impact on the LLM community.  We address your questions below.
>
> # Comment 1.1: "The major contribution of this paper is to propose the problem of RLHF in a continual manner, however, the motivation and necessity of this problem is lack of detailed explanation and support, hindering readers’ understanding of its value.
>
> * Response: The motivation of  Continual RLHF: In real-world applications, learning continuously changing human preferences is more practical than learning invariable human preferences.
> For example, the progression from the onset of the COVID-19 virus in human society to widespread infections and finally to achieving herd immunity has seen corresponding changes in government policies and human perspective.
> An AI agent that keeps pace with the times should exhibit behavior that aligns with current government policies and human understanding preferences at different stages, rather than remaining static.
>
> However, traditional alignment methods, namely Reinforcement Learning from Human Feedback (RLHF)[1,2], lack flexibility for continual learning (CL) of human preferences. Existing work[3] often necessitates retraining models to adapt to dynamic preferences, which needs huge computation resources. Hence, there is a pressing need for research into continual alignment methods to address this limitation, enabling LLMs to better adhere to evolving human preferences and values while generating helpful responses. We note that reviewers za1s and Loz4 mentioned the motivation of CPPO under Strengths as well.
>
> # Comment 1.2:  "The authors should introduce more related work that retrain or finetune the LM when new data arrive. "
>
> * Response:  According to your suggestion, we have added more related work in Section 5.2 of the new PDF.
>
> # Comment 1.3: " Furthermore, the storage and computational cost of different approaches should be analyzed, for example, the retraining methods, replay-based methods and regularization-based methods."
>
> * Response: The memory and computational complexity of the continual learning methods depend on the actual experiment setting. In general, the storage complexity of Replay-based methods is positively correlated with the number of replayed samples and linearly increases with the number of tasks. Regularization-based methods, such as EWC, typically require storing the weights of historical models and Fisher matrix information, leading to storage costs that linearly increase with the number of tasks. On the other hand, Re-training methods necessitate retraining the entire historical dataset, resulting in cumulative computational costs being quadratic with respect to the number of tasks.
>
> The methods in Section 5.2 have been applied in NLP scenarios for continual pre-training, task incremental learning, class incremental learning, and domain incremental learning[4]. To the best of our knowledge, these methods have not been experimentally validated in the continual learning of the RLHF scenario.
>
>
>
>
> # References
> [1] Nisan Stiennon, et al.  Learning to summarize from human feedback, arxiv 2020.
>
> [2] Long Ouyang, et al. Training language models to follow instructions with human feedback, arxiv 2022.
>
> [3] Yuntao Bai, et al. Training a helpful and harmless assistant with reinforcement learning from human feedback. arxiv 2022.
>
> [4] Magdalena Biesialska, et.al. International Committee on Computational Linguistics. 2020.

---

> ### Author Response · Authors · 2023-11-20
> **Response to Reviewer CUAu**
>
> # Comment 2.1: "The experiment part is weak.  In the main experiment to evaluate the continual learning ability of CPPO in NLP tasks, only one benchmark Reddit TL;DR summarization is considered. What’s more, the length of the continual task stream is only 2, which is not enough for a solid evaluation of continual learning ability compared to related work... "
>
> * Response: To further validate the proposed method, we add a new benchmark for Domain-Incremental Learning (DIL) **on 18 domains, comprising 3 tasks**.
> We build this benchmark based on the Stanford Human Preferences Dataset (SHP)[1] dataset. Our task design methodology can also be extended to sequences of 6, 9, or even 18 tasks.
> Here, we only present the final experimental results. The details of the SHP dataset, task design, and evaluation metrics can be referenced in the **new PDF Appendix Section H**.
>
>
> |                                   | Task 1                               | Task 2                               | Task 3                      |                                     |                                      |                                       |                                      |   |
> |-----------------------------------|--------------------------------------|--------------------------------------|--------------------------------------|-------------------------------------|--------------------------------------|---------------------------------------|--------------------------------------|---|
> | **Method** | **SteamSHP (↑)**                | **SteamSHP (↑)**                | **SteamSHP (↑)**                | **AA (↑)**                     | **AIA (↑)**                     | **BWT (↑)**                      | **FM (↓)**                      |
> |       SFT (In order)    | 0.806 $(\pm0.0101)$          | 0.836 $(\pm0.0103)$          | 0.853 $(\pm0.0103)$          | 0.832 $(\pm0.0061)$         | 0.837 $(\pm0.0039)$          | -0.022 $(\pm0.0094)$          | 0.022 $(\pm0.0094)$          |
> |    SFT (multi-tasks)      | 0.831 $(\pm0.0266)$          | 0.847 $(\pm0.0145)$          | 0.858 $(\pm0.0114)$          | 0.845 $(\pm0.0147)$         | 0.844 $(\pm0.0082)$          | -0.006 $(\pm0.0183)$          | 0.009$(\pm0.0160)$           |
> |   Iterated RLHF   | 0.869 $(\pm0.0583)$          | 0.88 $(\pm0.0490)$           | 0.887 $(\pm0.0421)$          | 0.879 $(\pm0.0488)$         | 0.874 $(\pm0.0433)$          | **0.0004** $(\pm0.0186)$ | **0.003** $(\pm0.0162)$ |
> |PPO| 0.853 $(\pm0.0573)$          | 0.879 $(\pm0.0428)$          | 0.889 $(\pm0.0369)$          | 0.874 $(\pm0.0433)$         | 0.877 $(\pm0.0378)$          | -0.017 $(\pm0.0351)$          | 0.020 $(\pm0.0327)$          |
> |CPPO(Heuristic)| 0.864 $(\pm0.0557)$          | 0.89 $(\pm0.0448)$           | 0.894 $(\pm0.0350)$          | 0.883 $(\pm0.0429)$         | 0.887 $(\pm0.0391)$          | -0.015 $(\pm0.0321)$          | 0.018 $(\pm0.0300)$          |
> | CPPO(Learn)                       | **0.872** $(\pm0.0544)$ | **0.898** $(\pm0.0450)$ | **0.899** $(\pm0.0342)$ | **0.89** $(\pm0.0424)$ | **0.894** $(\pm0.0389)$ | -0.013 $(\pm0.0298)$          | 0.016 $(\pm0.0281)$          |
>
>
> # Comment 2.2: "The proposed CPPO method can also be applied to other continual reinforcement learning tasks other than RLHF. Simpler conventional control tasks like DoorGym should be included to further validate the soundness of CPPO."
>
> * Response: We believe that the evaluation of DoorGym is out of the scope of our paper. Our research focuses on the alignment of LM, following previous works [2-6] which are evaluated only on NLP tasks.
> In this paper, we have evaluated CPPO on 5 representative NLP datasets, including Reddit TL;DR for summarization, CNN Daily Mail for summarization, IMDB for sentiment text generation, Random Walks for generating the shortest path, and SHP for open-domain QA.
>
> # References
> [1] Kawin Ethayarajh, et al. Understanding dataset difficulty with V-usable information. PMLR, 2022.
>
> [2] Yuntao Bai, et al.  Training a helpful and harmless assistant with reinforcement learning from human feedback. arxiv, 2022.
>
> [3] Yuntao Bai, et al.  Constitutional ai: Harmlessness from ai feedback. arxiv, 2022.
>
> [4] Rafael Rafailov, et al. Your language model is secretly a reward model, NeurIPS, 2023
>
> [5] Nisan Stiennon, et al. Learning to summarize from human feedback. arxiv, 2020.
>
> [6] Zheng Yuan, et al. Rrhf: Rank responses to align language models with human feedback without tear, NeurIPS, 2023

---

> > ### Comment · Reviewer_CUAu · 2023-11-21
> >
> > Thanks for your reply! My concerns have been addressed, and I raise my score to 6.

---

> > > ### Author Response · Authors · 2023-11-22
> > > **Response to Reviewer CUAu**
> > >
> > > Thank you again for your helpful review, and we are glad that we addressed your concerns.  **Please also let us know if you have anything else to ask**, because this would help really us improve the work. Finally, we sincerely appreciate you raising your score.

---

### Official Review · Reviewer_za1s · 2023-10-31

**Soundness:** 4 excellent
**Presentation:** 4 excellent
**Contribution:** 3 good
**Rating:** 8
**Confidence:** 2

**Summary:**

The paper proposes a new method, CPPO, to align language models with human preferences in the continual learning setting, exempting the need to do LM retraining. Based on PPO, CPPO introduces a new imitation loss term to imitate the old policy outputs, and a weighting strategy is applied to balance the policy gradient loss and the imitation loss. The weighting strategy can be implemented in a heuristic way or a learnable way through the Lagrange function. As a result, in the continual learning setting, the policy (Language model) can seek a good trade-off between new task performance and old knowledge retention. The experiments show that the method outperforms all baselines and is close to the ChatGPT in terms of several metrics such as reference preference model score.

**Strengths:**

1. The work aims to address a practical and important issue, language model alignment in a dynamic human preference setting.
2. The paper is well-written.
3. The main idea and the motivation behind it make sense.
4. The evaluation is thorough with reasonable metrics and adequate baselines.

**Weaknesses:**

I wonder about the performance of 1) $M_{\pi2}$ on the Task-1 test set and 2) $M_{\pi1}$ on the Task-2 test set. The result of the first experiment can show how much knowledge the model retains for the first task. In addition, the second experiment is supposed to show a mediocre result to prove the two tasks have a clear difference and that the experiment setting is indeed a continual learning setting. The performance gap of $M_{\pi1}$ and $M_{\pi2}$ on Task-2 can reveal to what extent the two data distributions mismatch.

**Questions:**

N/A

---

> ### Author Response · Authors · 2023-11-20
> **Response to Reviewer za1s**
>
> Thank you for your helpful comments! We appreciate your comments about our main idea and the motivation behind it make sense. We address your questions below.
>
> # Comment:  "I wonder about the performance of 1) $M_{\pi_2}$ on the Task-1 test set and 2) $M_{\pi_1}$ on the Task-2 test set. The result of the first experiment can show how much knowledge the model retains for the first task. In addition, the second experiment is supposed to show a mediocre result to prove the two tasks have a clear difference and that the experiment setting is indeed a continual learning setting. The performance gap of $M_{\pi_1}$ and $M_{\pi_2}$ on Task-2 can reveal to what extent the two data distributions mismatch."
>
> Response: The reference PM score of  $M_{\pi_2}$ on Task 1 can be computed by $rPM(M_{2}, \mathbb{D}_1^{test} )=rPMS_1 - SFR$.
>
> Here, we supplement the performance of $M_{\pi_1}$ on Task 2, comparing CPPO with PPO.
>
>
> | Method   | $rPM(M_{1}, \mathbb{D}_1^{test} )$ | $rPM(M_{1}, \mathbb{D}_2^{test} )$ | $rPM(M_{2}, \mathbb{D}_1^{test} )$ | $rPM(M_{2}, \mathbb{D}_2^{test} )$ |
> |----------|------------------------------------|------------------------------------|------------------------------------|------------------------------------|
> | PPO      | 2.631                              | 2.417                              | 2.550                              | 2.688                              |
> | CPPO (H) | 3.021                              | 2.812                              | 3.187                              | 2.982                              |
> | CPPO (L) | 3.174                              | 2.714                              | 3.337                              | 3.090                              |
>
> Comparing the evaluations in columns 2 and 4 for $M_{\pi_1}$ and $M_{\pi_2}$ on Task-2, we observe that model $M_{\pi_1}$  significantly underperforms $M_{\pi_2}$. This indicates a substantial difference in the data distributions between the two tasks.
>
> Comparing the evaluations in columns 1 and 3 for $M_{\pi_1}$ and $M_{\pi_2}$ on Task-1, we observe that the policy model trained with PPO exhibits forgetting, indicating a risk of catastrophic forgetting during continually learning of Task-1 and Task-2. The model trained with CPPO demonstrates Backward Transfer (BWT) capability, specifically a negative SFR, suggesting that CPPO overcomes forgetting in this task and that learning new human preferences leads to a positive impact on retaining old preferences.

---

> > ### Comment · Reviewer_za1s · 2023-11-21
> > **Thank you**
> >
> > Thank you for the response.
> >
> > I am glad to see the new evaluation results. The performance drop of $PPO \ M_2$ on dataset #1 emphasizes the necessity of this method in the continual learning setting.
> >
> > I am mainly with RL application background. From my perspective, the modifications based on PPO are common and make sense. Also, the quantitative results are good. The only thing I am not sure about is whether the evaluation metrics are widely accepted by the NLP community to evaluate a generative model. I guess it is a sticky question for the community as well, as the best way is through human evolution, which is costly. As the $rPM$ is trained on the entire dataset and with many more parameters than the LM, I personally accept that it is able to serve as a reasonable proxy to evaluate the language model. Thus I still believe the technique and result of this paper are sound and would like to keep my score unchanged.

---

> > > ### Author Response · Authors · 2023-11-22
> > > **Response to Reviewer za1s**
> > >
> > > Thank you again for your helpful review. We appreciate your comments that the modifications based on PPO are common and make sense, and that the technique and results of this paper are sound.
> > > We address the evaluation metric issue you are concerned below.
> > >
> > >
> > > As you pointed out, the evaluation metric of evaluate a LLM indeed is a sticky question for the community.
> > > In our work, we use three types of metrics for evaluation, including 1) the N-gram overlap metric, Rouge, 2) task-specific, model-based human preference metrics, rPMS, 3) human evaluation using the Likert score.
> > >
> > > Rouge is the most widely used metric in summarization tasks [1]. Using a large reward model as the gold reward model follows recent research on Reinforcement Learning from Human Feedback (RLHF) by OpenAI [2]. The Likert score is also widely used to evaluate LLMs, and influential works in NLP, such as InstructGPT [3], utilize the Likert score as an evaluation metric.
> > >
> > > Hence, we believe that the metrics used are widely accepted by the NLP community.
> > >
> > > ## Thank you
> > >
> > > Thank you for your feedback. **Please let us know whether we have addressed your concerns**.
> > > If you find that we have properly addressed your concerns, we kindly request that you consider adjusting your **confidence** score accordingly.
> > >
> > >
> > > ## References
> > >
> > > [1] Stiennon, Nisan, et al. "Learning to summarize with human feedback." Advances in Neural Information Processing Systems 33 (2020): 3008-3021.
> > >
> > > [2] Gao, Leo, John Schulman, and Jacob Hilton. "Scaling laws for reward model overoptimization." International Conference on Machine Learning. PMLR, 2023.
> > >
> > > [3] Ouyang, Long, et al. "Training language models to follow instructions with human feedback." Advances in Neural Information Processing Systems 35 (2022): 27730-27744.

---

> > > > ### Comment · Reviewer_za1s · 2023-11-22
> > > > **Response**
> > > >
> > > > My expertise mainly lies in reinforcement learning. I am confident about the RL-related part of this submission. The technical motivation makes sense and I am not surprised it works well.
> > > >
> > > > However, I am with limited  NLP background and thus can not provide a higher confidence in terms of the evaluation part. Sorry about that. But as I said before, I am inclined to believe that the evaluation is reasonable in terms of the language/reward model design and the metrics.

---

### Official Review · Reviewer_Loz4 · 2023-10-31

**Soundness:** 2 fair
**Presentation:** 2 fair
**Contribution:** 2 fair
**Rating:** 5
**Confidence:** 4

**Summary:**

This paper looks at a more ambitious form of RLHF that does not require complete retraining from scratch when new feedback is introduced as a result of human preferences that may vary over time across domains and topics. The authors propose to modify the PPO algorithm leveraged within RLHF training by designing an algorithm to accomplish 4 key desiderata:

1. Examples with high reward and high generation probability have a high policy learning weight and high knowledge retention weight.
2. Examples with high reward and low generation probability or high generation probability and low reward have a high policy learning weight and low knowledge retention weight.
3. Examples with low reward and low generation probability have a low policy learning weight and low knowledge retention weight.
4. All other examples are learned with default weightings.

The authors then conduct experiments comparing their proposed method favorably during continual RLHF training to generic PPO as well as relevant baselines and ablations.

**Strengths:**

- I like the central motivation of this paper as I believe it is an important setting that has been under-explored to date in the context of RLHF.
- The experimental results seem pretty good, suggesting that there may at least be some strong insights coming out of this paper about the impact of extreme examples of the efficacy of continual training.
- Focusing on different treatment of extreme examples is a bit outside the norm within the continual learning context that is generally focused on either experience replay or knowledge retention strategies coming from different motivations. This gives the paper a certain degree of originality, but I feel it also makes justification of the particular approach more important, which is an area where I find the paper to be lacking.
- There are a number of useful charts and tables throughout the text that help readers get the main idea when they may get confused.

**Weaknesses:**

- The continual RL problem solved by this paper is never really formulated. Equation 4 is proposed as an objective, but it is never made clear what problem it solves. This makes it feel a bit arbitrary to me. For example, see recent work proposing formal definitions of the problem [1], [2].
- In general I find section 3 prior to section 3.2.1 to be very hard to follow. A number of things are referred to as theoretically justified or derived, but as far as I can tell this is not true in any meaningful sense. I think I would be much more open to the positioning of the contribution if the authors just started with the desiderata of section 3.2.1 and explained the reason for each intuitively maybe with the aid of a little bit of math for each one (something that is currently lacking in my view). The paper could be better positioned in my view more as an exploration of the impact of these various intuitions and their impact on continual RLHF. This is because, in my mind, the current discourse does not really deliver on proposing this technique as a theoretically justified approach for RL.
- Improvements in Table 6, Fig 5, and Table 8 seem to be there but it is not clear how significant these are or exactly why this would generalize across domains. I didn’t notice any analysis of statistical significance, which is odd because PPO (and most RL methods) are known to have high run to run variance in general, so it is very hard to take RL results seriously without this.

Because I worry about statistical significance and generality of the results and don't believe the formulation is well justified as presented, I lean towards rejection at this time.

[1] Khetarpal, Khimya, et al. "Towards continual reinforcement learning: A review and perspectives." Journal of Artificial Intelligence Research 75 (2022): 1401-1476.

[2] Kumar, Saurabh, et al. "Continual learning as computationally constrained reinforcement learning." arXiv preprint arXiv:2307.04345 (2023).

Detailed Comments on Equation 5:

A particular issue with the start of section 3 is with equation 5 and the discussion around it. What do you mean when you say “By introducing the actor-critic version, the clipped ratio, and the entropy bonus, we claim that Eq.(4) can be improved to (the derivation is detailed in Appendix Section B)”? What does "improved" mean in this context? Also going through Appendix Section B, it seems apparent to me that this is not a derivation of any particular theory/lemma/proposition or fact, rather it is just a series of steps that are not explained in the main text. For example, the authors mention improvement when they say: “In the PPO method, the objective is to maximize the expectation of the advantage function instead of the reward value. Hence, we improve the above objective as” -- here the "improvement" is either empirical or related to the theory about bias/variance. It is mentioned again when they say: “we introduce the knowledge retention penalty instead of the true KL divergence, we discuss the reason in lines 134-137 in our paper. Here, the above objective is improved as:”  -- here the improvement is based on the authors own empirical observations and computational justification. The authors also write "Then we introduce the importance sampling like PPO, the above objective can be written as" -- importance sampling is a deep theoretical topic related to off-policy optimization while having issues related to variance to implement in practice. I find the analysis of and flippant discussion of this component of the algorithm to be entirely inadequate in the appendix. It is also not even discussed in the main text.

**Questions:**

1. What is the statistical significance of the reported results across random seeds?
2. What problem formulation is equation 4 a solution to?
3. The key difference with generic RLHF seems to be that the optimization of the two terms is only over a subset of the data in equation 4. Why does it help us to essentially throw away data? Or is this motivated as part of a computational constraint from some implicitly considered problem formulation that is not spelled out?
4. The hard cutoff based on hyperparameter k seems a bit weird to me. Towards what metric would the hyperparameter k be optimized for if it can't be equation 4 itself? This gets even weirder for me when the hard cutoff is then relaxed in equation 6. I don't get why it was ever introduced to begin with.
5. Why does equation 6 make equation 5 easier to optimize? I buy that equation 6 is more general than equation 5, but this is the key motivation for this sense and it is really not clear to me why this would be the case. Especially considering that focusing on only a subset of the data would require less computation, which I presume is a major constraint of the implicit problem formulation.

---

> ### Author Response · Authors · 2023-11-20
> **Response to Reviewer Loz4**
>
> Thank you for your helpful comments! We appreciate your positive feedback on our research motivation and experimental results.  We address your questions below.
>
> # Comment 1: "What is the statistical significance of the reported results across random seeds?"
> * Response: Due to time and computational constraints, we currently do not have the statistical significance results for all the baselines. In subsequent versions of the paper, we will include the corresponding experimental results.  In the table below, we present the mean and variance of the evaluation results for some methods across three random seeds (due to the randomness, there may be slight variations compared to the original PDF).
>
>
>
> | Method                             | **rPMS$_1$ ($\uparrow$)**    | **rouge ($\uparrow$)**       | **AT ($\downarrow$)**        | **rPMS$_2$ ($\uparrow$)**    | **rouge ($\uparrow$)**       | **SFR ($\downarrow$)**        | **rPMS ($\uparrow$)**        | **rouge ($\uparrow$)**       |
> |------------------------------------|-----------------------------------|-----------------------------------|-----------------------------------|-----------------------------------|-----------------------------------|------------------------------------|-----------------------------------|-----------------------------------|
> | **SFT (In order)**            | 1.499 $\pm$0.130          | **0.248** $\pm$0.006 | $-$                               | 1.543 $\pm$0.067          | **0.237** $\pm$0.007 | $-$                                | 1.498 $\pm$0.051          | **0.237** $\pm$0.009 |
> | **SFT (multi-tasks)**         | 1.524 $\pm$0.041          | 0.254 $\pm$0.011          | $-$                               | 1.536 $\pm$0.092          | 0.234 $\pm$0.009          | $-$                                | 1.505 $\pm$0.011          | 0.236 $\pm$0.008          |
> | **PPO (In order)**        | 2.629 $\pm$0.183          | 0.196 $\pm$0.050          | 0.052 $\pm$0.044      | 2.546 $\pm$0.201          | 0.151 $\pm$0.022          | 0.144 $\pm$0.024          | 2.502 $\pm$0.242          | 0.186 $\pm$0.016          |
> | **Iterated RLHF**         | 2.629 $\pm$0.183          | 0.196 $\pm$0.050          | 0.052 $\pm$0.044           | 2.732 $\pm$0.163          | 0.211 $\pm$0.011          | 0.061 $\pm$0.018          | 2.666 $\pm$0.124          | 0.200 $\pm$0.010          |
> | **PPO**             | 2.629 $\pm$0.183          | 0.196 $\pm$0.050          | 0.052 $\pm$0.044          | 2.687 $\pm$0.126          | 0.184 $\pm$0.017          | 0.080 $\pm$0.017          | 2.612 $\pm$0.191          | 0.188 $\pm$0.013          |
> | **PPO+OnlineL2 Reg** | 2.758 $\pm$0.121          | 0.206 $\pm$0.042          | 0.042 $\pm$0.042         | 2.701 $\pm$0.205          | 0.180 $\pm$0.012          | 0.062 $\pm$0.013          | 2.700 $\pm$0.114          | 0.196 $\pm$0.011          |
> |   **PPO+EWC**    | 2.833 $\pm$0.122          | 0.201 $\pm$0.043          | 0.047 $\pm$0.039   |     2.823 $\pm$0.192          | 0.175 $\pm$0.022          | 0.040 $\pm$0.015          | 2.801 $\pm$0.202          | 0.196 $\pm$0.023          |
> | **HN-PPO**    | 2.859 $\pm$0.051          | 0.212 $\pm$0.034          | 0.036 $\pm$0.042       | 2.868 $\pm$0.132          | 0.171 $\pm$0.017          | 0.028 $\pm$0.029          | 2.846 $\pm$0.177          | 0.201 $\pm$0.011          |
> |**CPPO (Heuristic)** | 3.020 $\pm$0.137          | 0.213 $\pm$0.024          | 0.035 $\pm$0.023          | 2.978 $\pm$0.113          | 0.174 $\pm$0.019          | -0.164 $\pm$0.009          | 3.099 $\pm$0.153          | 0.179 $\pm$0.016          |
> | **CPPO (Learn)**     | **3.180** $\pm$0.154 | 0.220 $\pm$0.040          | **0.028** $\pm$0.042 | **3.085** $\pm$0.134 | 0.164 $\pm$0.024          | **-0.161** $\pm$0.008 | **3.207** $\pm$0.113 | 0.179 $\pm$0.008          |

---

> ### Author Response · Authors · 2023-11-20
> **Response to Reviewer Loz4**
>
> # Comment 2: "What problem formulation is equation 4 a solution to?"
>
> * Response: In the continual RLHF scenario, the human preference dataset evolves over time, often consisting of a sequence of tasks or domains. Each task or domain may have its own set of data, and these tasks are presented to the model sequentially. The model needs to adapt to new tasks without forgetting previously learned ones.
>
> * **Task Formulation**:  In this paper, we propose the task of continually learning human preferences under an offline continual learning setting[1].
> Formally. we consider a task sequence of  $T = {T_1 \rightarrow T_2 \rightarrow  ... }$ to continually learn a policy model on the corresponding human preference datasets  $HF =\{HF_1 \rightarrow  HF_2 \rightarrow ...\}$ and prompt  datasets  $S =\{S_1 \rightarrow
>  S_2 \rightarrow  ...\}$.  For each task $T_t (t=1,2,...)$,  the policy  $\pi_t$ is initialized by $\pi_{t-1}$ and then is trained against the reward model $r_t$ , where the reward model $r_t$  is learned on $HF_t$.  The initial policy $\pi_0$ is the SFT model, namely, $\pi_0 = \pi_{SFT}$. The final objective is to learn a policy model $\pi_\theta$ that maximizes the overall reward on all of the learned human preferences:  $ max_{\theta}  \Sigma_{t=1}^{T} E_{s\sim{S_t}, a\sim \pi_{\theta}(\cdot \mid s)} \bigl[r_t(s,a)\bigr]$. **We also add the task formulation to the new PDF.**
>
> # Comment 3:  "The key difference with generic RLHF seems to be that the optimization of the two terms is only over a subset of the data in equation 4. Why does it help us to essentially throw away data? Or is this motivated as part of a computational constraint from some implicitly considered problem formulation that is not spelled out?  "
>
> * Response:
>
>  Eq.4 derives from the learning objective of the Policy Consolidation[2] method, as described in Eq.3 (in the new PDF):  $ \max_{\theta} E_{s\sim {S_t}, a\sim \pi_{\theta}(\cdot \mid s)} \bigl[r_t(s,a)\bigr] - E_{s \in {S_{t-1}}} D_{KL}({P_{\pi_{\theta}}} (a|s)||{P_{\pi_{t-1}}}(a|s))$.
>
> The relationship of $D_1$ in Eq.4 and $S_t$ in Eq.3 is $D_1 \subset S_t$, analogously  $D_2 \subset S_{t-1}$.
> Eq.4 is based on the intuitive objective: **to make the language model have a high probability of generating a response with a high reward.** Next, we clarify the reason that the objective Eq.4 is optimized on the subset of rollouts, not all of them.
>
> Due to the LM usually using the top-P or top-K sampling method to generate the response, namely responses with high generation probability are more potentially generated. Therefore, **the impact of low-probability samples on the final evaluation is weaker than the high-probability samples**.  Hence, we expect only to maximize the average reward on the rollout set with **high** generation probability in $S_t$, namely $D_1$.
>
> To prevent forgetting, we focus more on the high-reward samples, because we expect to **keep the ability of the old policy to generate high-reward rather than low-reward samples**.
> In our learning strategy, those responses with **low rewards can be forgotten when learning new preferences**. Hence, we minimize the KL divergence on the responses with high rewards in $S_{t-1}$, namely $D_2$, which is essentially a form of function regularization[1].
>
> The above intuition seeks a "compromise" between plasticity (learning new preference) and stability (retaining old preference).  It is similar to the EWC algorithm, where parameters with high importance are subjected to stronger regularization to keep the old knowledge, while parameters with low importance receive weaker regularization and have more capacity to learn new knowledge.
>
> # References:
>
> [1] Magdalena Biesialska, et al. Continual lifelong learning in natural language processing: A survey. ICCL, 2020.
>
> [2] Christos Kaplanis, et al. Policy consolidation for continual reinforcement learning. PMLR, 2019.

---

> ### Author Response · Authors · 2023-11-20
> **Response to Reviewer Loz4**
>
> # Comment 4: "The hard cutoff based on hyperparameter k seems a bit weird to me. Towards what metric would the hyperparameter k be optimized for if it can't be equation 4 itself? This gets even weirder for me when the hard cutoff is then relaxed in equation 6. I don't get why it was ever introduced to begin with."
>
> * Response:
> The introduction of $k$ is derived from the mean squared error (MSE) method for anomaly detection, aiming to identify different types of rollouts.
> In our scenario, if the generation probability or reward value deviates from its mean value $\mu$ exceeds $k$ times the standard deviation $\sigma$, namely $|p-\mu(p)|>k\cdot \sigma(p)$ or $|r-\mu(r)|>k\cdot \sigma(r)$, it is labeled as an anomaly, corresponding to a non-normal rollout type.
> Hence, hyperparameter $k$ controls the threshold of different rollout types.
> If we increase $k$, the size of the normal region in Figure 1 will increase, while the size of other regions will decrease.
> This means that the majority of rollout samples will learn like vanilla PPO.
>
>
> # Comment 5:  "Why does equation 6 make equation 5 easier to optimize? I buy that equation 6 is more general than equation 5, but this is the key motivation for this sense and it is really not clear to me why this would be the case. Especially considering that focusing on only a subset of the data would require less computation, which I presume is a major constraint of the implicit problem formulation."
>
> * Response:
> Since most rollout samples belong to the normal type (refer to Figure 1), optimizing using Eq.5 implies discarding a large portion of rollout samples, significantly reducing training efficiency (as the discarded samples also require time to generate). Therefore, for normal-type samples, we adopt a learning strategy similar to PPO, setting the weight $\alpha(x)$ to (or close to) 1. In this way, CPPO fully utilizes each rollout sample and adaptively adjusts the learning weights based on the reward and generation probability.

---

> ### Author Response · Authors · 2023-11-22
> **Response to Reviewer Loz4**
>
> We've taken your initial feedback into careful consideration and incorporated them into our manuscript as indicated in our response. Could you kindly confirm whether our responses have appropriately addressed your concerns? If you find that we have properly addressed your concerns, we kindly request that you consider adjusting your initial score accordingly. We understand that you are very busy, but would highly appreciate it if you could take into account our response when having discussions with AC and other reviewers.  **Please let us know if you have further comments**.
>
> Thank you again for your time and effort in reviewing our work.

---

> ### Author Response · Authors · 2023-11-23
> **Response to Reviewer Loz4**
>
> We noticed your concerns regarding Eq.5 in the **Weaknesses** part, and we guess that you may have questions about the derivation in Appendix B. Here, we will list out the derivation step by step, and if you have any specific doubts about a particular step, please feel free to point them out. And **we are eagerly waiting for your feedback on our response**.
>
> ## from Eq.4 to Eq.5 (same with the new PDF Appendix B)
>
> By introducing the actor-critic version, the clipped ratio, the objective Eq. 4. can be written as the objective Eq. 5. Next, we provide the derivation step by step:
>
>
> 1. We utilize notations $I_{D_1}(x)$ and $I_{D_2}(x)$ to rewrite the Eq.4 as $\max_{\theta} \mathbb E_{x \sim \pi_{\theta}} I_{D_1}(x) \cdot R(x) - \mathbb E_{x \sim \pi_{t-1}} I_{D_2}(x) \cdot D_{KL}({P_{\pi_{\theta}}} (x)\parallel P_{\pi_{t-1}}(x))$.
>
> 2. Then we introduce the importance sampling like PPO, the above objective can be written as $\max_{\theta} \mathbb E_{x \sim \pi_{t-1}} I_{D_1}(x) \cdot \frac{{P_{\pi_{\theta}}} (x)}{{P_{\pi_{t-1}}} (x)} {R}(x) - \mathbb E_{x \sim \pi_{t-1}} I_{D_2}(x) \cdot D_{KL}({P_{\pi_{\theta}}} (x)\parallel{P_{\pi_{t-1}}}(x))$.
>
>
> 3. In the PPO method, the objective is to maximize the expectation of the advantage function instead of the reward value. By introducing the advantage function instead of the reward, the above objective can be written as $\max_{\theta} \mathbb E_{x \sim \pi_{t-1}} I_{D_1}(x) \cdot \frac{{P_{\pi_{\theta}}} (x)}{{P_{\pi_{t-1}}} (x)} {A}(x) - \mathbb E_{x \sim \pi_{t-1}} I_{D_2}(x) \cdot D_{KL}({P_{\pi_{\theta}}} (x)\parallel{P_{\pi_{t-1}}}(x))$.
>
> 4. By introducing the knowledge retention penalty instead of KL-divergence, the above objective is written as: $\max_{\theta} \mathbb E_{x \sim \pi_{t-1}} I_{D_1}(x) \cdot \frac{{P_{\pi_{\theta}}} (x)}{{P_{\pi_{t-1}}} (x)} {A}(x) - \mathbb E_{x \sim \pi_{t-1}} I_{D_2}(x) \cdot L^{KR}(x)$.
>
>
> 5. In the CL task, the new policy $\pi_{t}$ is generally initialized by the old policy $\pi_{t-1}$. In CPPO, we treat the $\pi_{t-1}$ and $\pi_{t}$ as the reference model and policy model respectively. Then, we consider the actor-critic version, the clipped ratio, and the entropy bonus used in PPO, the above objective can be written as ${J}({\theta})^{'} = L^{ {I_{D_1}} \cdot  CLIP+ {I_{D_2}} \cdot KR + VF + S}(\theta)$, namely the Eq. 5.

---

### Official Review · Reviewer_qeM7 · 2023-11-01

**Soundness:** 2 fair
**Presentation:** 3 good
**Contribution:** 2 fair
**Rating:** 6
**Confidence:** 2

**Summary:**

This work studies the problem of continual learning from humans, where the signal from humans takes the form of a preference. The main challenge of continual learning is retaining past knowledge, while still able to maximize reward on the newly acquired preferences. To stabilize the training procedure, this work proposes "stable learning", favoring actions that are both a) high probability under the existing model -- i.e. retaining past knowledge and b) high reward for the new preferences acquired -- i.e. conforming to the new preferences.

These two aspects are turned into "knobs" \alpha and \beta, which are used to weigh parts of the objective function to either increase the effects of learning new knowledge or retaining old knowledge. The values of \alpha and \beta are adjusted by collecting roll-out samples and classifying them into 5 categories, normal, high performing, ... noise. Where each category has an influence on the adjustments on these two knobs.

Empirical results are promising.

**Strengths:**

## originality : fair
The paper proposes a straight-forward (but nonetheless novel) idea to use samples in the continual learning process to modify the learning rates, whereby adjusting the rate in which the model retains old knowledge and learns new knowledge.

## Clarity : good
The approach is clearly explained.

**Weaknesses:**

## Quality : less than ideal.
It is unclear if the proposed method is actually better than the baselines from table 8, as there is no confidence intervals being computed, nor is there statistical tests being performed. Depending on the results, the authors might have to conduct additional human evaluations, so that the confidence intervals "pulls apart".

Table 8 is the only evaluation that is conducted directly against humans, it is the only "non-proxy" evaluation of the proposed method -- directly asking humans to rate the responses -- rather than evaluating it indirectly through a reward model.

As such, the authors should look to conduct evaluations more in the form of direct human ratings, to give readers who are not too familiar with the details of RLHF (such as myself) confidence that the approach works well "end to end", when evaluated directly against humans.

## Significance : unclear
As someone not directly in the RLHF community, I leave this to other seasoned reviewers.

**Questions:**

please provide confidence intervals and t-tests for table 8. It would be good to show the proposed method is significantly better than PPO.

---

> ### Author Response · Authors · 2023-11-20
> **Response to Reviewer qeM7**
>
> Thank you for your helpful comments! We appreciate your comments that our idea is novel and our approach is clearly explained.  We address your questions below.
>
> # Comment 1: "Please provide confidence intervals and t-tests for table 8. It would be good to show the proposed method is significantly better than PPO."
>
> * Response: We conduct t-tests and compute confidence intervals and p-values for Table 8 in the original paper. The detailed results are shown in the table below. In particular, the proposed method CPPO improves the PPO baseline by 8.23\% with a p-value of 0.037, indicating that CPPO is significantly better than PPO.
>
> | Method  | Likert score$\pm$std| Improve | 0.95 CI        | P-value  |
> |---------|------------------------------|---------|----------------|----------|
> | PPO     | 4.370$\pm$1.180      | -       | (4.134, 4.605) | -        |
> | CPPO    | 4.730$\pm$1.231      | 8.23\%  | (4.484, 4.976) | 0.037    |
> | ChatGPT | 4.760$\pm$1.011      | 8.92\%  | (4.558, 4.962) | 0.013    |
> | Human   | 4.900$\pm$1.034      | 12.13\% | (4.694, 5.106) | 0.001    |

---

> > ### Comment · Reviewer_qeM7 · 2023-11-23
> > **thanks for running these tests**
> >
> > this sufficiently addresses the quality concern that I had. raising score to 6.

---

> ### Author Response · Authors · 2023-11-22
> **Response to Reviewer qeM7**
>
> We are grateful for your time and efforts. We have been eagerly waiting for your feedback on our response. We will be here waiting and hope to see it before the discussion period ends. We understand that you are very busy, but would highly appreciate it if you could take into account our response when updating the rating and having discussions with AC and other reviewers. **Please let us know if you have further comments**.
>
> Considering that you may not be familiar with the RLHF field, we provide some RLHF references[1-3] that are highly relevant to our study. We hope that these works will assist you in understanding and evaluating the research content of our paper.
>
>
> ## References
>
> [1] Stiennon, Nisan, et al. "Learning to summarize with human feedback." Advances in Neural Information Processing Systems 33 (2020): 3008-3021.
>
> [2] Ouyang, Long, et al. "Training language models to follow instructions with human feedback." Advances in Neural Information Processing Systems 35 (2022): 27730-27744.
>
> [3] Bai, Yuntao, et al. "Training a helpful and harmless assistant with reinforcement learning from human feedback." arXiv preprint arXiv:2204.05862 (2022).

---

### Author Response · Authors · 2023-11-20
**General Response**

We are grateful to all reviewers for their insightful comments. We appreciate that reviewers found CPPO to be novel and original (qeM7, Loz4) / well-motivated  (Loz4, za1s) / reasonable metrics and evaluation (za1s) /  addressing a practical and important issue (za1s) / compared with adequate baselines (za1s, CUAu) / well-written (za1s, CUAu) / clearly explained  (qeM7) / having a great impact to the LLM community (CUAu).}

According to your suggestions, we have now uploaded a new PDF which adds:
1. the detailed explanation of the motivation for this study (Section 1),

2. the task formulation of the continual RLHF (Section 2),

3. more related works about continually training LM (Section 5.2),

4. new continual RLHF benchmark with 3 task steam across 18 domains and new experiments on Llama-7b (Appendix Section H).

---

> ### Author Response · Authors · 2023-11-21
> **Please let us know if you have any further questions!**
>
> We thank all reviewers again for their insightful comments and questions.
>
> We have attempted to carefully address all of your questions in the individual responses and in the edited paper draft. If you have any additional questions during the last day of the discussion period, we would be delighted to provide further clarification.

---

### Public Comment · ~Jiahao_Zhao1 · 2024-12-09
**Great Work! Any plan to release a repo?**

The idea of balancing policy learning and knowledge retention with sample-wise weights is both innovative and solid.

I noticed the supplementary material includes some code, but it seems incomplete. Would there be any plans to release a full repository? Access to the complete code would greatly benefit the research community and allow for further exploration of your promising approach.

Looking forward to hearing from you.

---

> ### Public Comment · ~Han_Zhang3 · 2024-12-10
> **Response**
>
> Thank you for your attention to our work. The code is complete. The new task continues training based on the checkpoint saved from the previous task, so we have only provided the code for the first task's training.

---

> > ### Public Comment · ~Jiahao_Zhao1 · 2024-12-16
> > **Thanks for Resposne**
> >
> > Thanks for your response. The codes in supplementary material are missing environment requirements and dependency package versions. It would be helpful if you could add environment setup instructions to reproduce the results in the paper.
> > Thanks in advance.

---

> > > ### Public Comment · ~Han_Zhang3 · 2024-12-16
> > > **Response**
> > >
> > > We use the trlx: v0.2 version which is released on Oct 22, 2022. You can directly follow this repo:   https://github.com/CarperAI/trlx/releases/tag/v0.2

---

> > > > ### Public Comment · ~Jiahao_Zhao1 · 2024-12-16
> > > > **Thanks for quick resposne!**
> > > >
> > > > If I understand correctly, the supplementary material has a local `trlx` that implemented CPPO, so installing the `trlx` package separately should not be necessary. However, it would be highly beneficial to provide a comprehensive list of dependencies, such as a full requirements.txt file or a Docker configuration, to streamline reproducibility.
> > > >
> > > > Additionally, there are **plenty of missing configurations** in the requirements. For example, the file `Experiments/continual-learning/summarize_rlhf/configs/cppoH_config_task-1.yml` lacks crucial parameters like `abl_type`, `threshold`, `ub`, and `lb`. While these can be found in Appendix E.1, this approach is highly inconvenient for reproducing the results.
> > > >
> > > > It seems that the code in the supplementary material is intended as a temporary solution, but it consists of plenty of dummy and unclear code. **The presentation of this paper is excellent, but the current state of the released code falls significantly below the standard expected at ICLR.** I strongly suggest providing a well-maintained public repository to enable others to reproduce your results more easily. This would greatly enhance the paper's impact and influence within the research community.

---

> ### Public Comment · ~Han_Zhang3 · 2024-12-16
> **About dependencies**
>
> Thanks for your comments.
> Because we are directly modifying the trlx code when implementing CPPO, it is consistent with trlx's dependency environment.
> This is the repo of CPPO: https://git.openi.org.cn/Hanlard/CPPO
> It includes the **requirements.txt** file of dependencies. We hope this is helpful to you.

---

> ### Public Comment · ~Han_Zhang3 · 2024-12-16
> **About configurations**
>
> CPPO has two versions, namely, CPPO (Heuristic) and CPPO (Learn).
> The CPPO (Learn) uses the learnable balance weights, hence it **does not need** the hyperparameters like ub, and lb.

---

### Meta-Review · Area_Chair_xt8j · 2023-12-10

**Metareview:**

This paper proposes a method for doing RLHF in continual settings. As Reviewer qeM7 says in their review: The main challenge of continual learning is retaining past knowledge, while still able to maximize reward on the newly acquired preferences. To stabilize the training procedure, this work proposes "stable learning", favoring actions that are both a) high probability under the existing model -- i.e. retaining past knowledge and b) high reward for the new preferences acquired -- i.e. conforming to the new preferences.

Overall, this paper is borderline. While on the one hand, the results of the paper are significant statistically and show that their approach improves over the prior methods or other continual learning approaches, it remains unclear if a method with this much complication is needed. The intuitions to derive the heuristic weighting scheme are not quite rigorous -- for example, "overfitting sample" does not refer to the classical definition of overfitting, etc. I am also unclear if from a continual learning perspective most recent continual learning approaches have been tested -- for example, adapting simply a constant weigthing across all datapoints by annealing it over time.

Despite these reasons, the paper provides some useful pointers for future works to build on. It would be useful to understand the benefits of this approach in toy settings and also understand if this approach can be made more rigorous or principled without needing several intuitive yet non-rigorous definitions.

**Justification For Why Not Higher Score:**

I think the paper has interesting results, but appears very hacky with several knobs. I am also not sure if all simple baselines have been compared to.

**Justification For Why Not Lower Score:**

N/A

---

### Decision · Program_Chairs · 2024-01-16

Accept (poster)